# The catecholamine precursor Tyrosine reduces autonomic arousal and decreases decision thresholds in reinforcement learning and temporal discounting

David Mathar[1]*, Mani Erfanian Abdoust[2], Tobias Marrenbach[2], Deniz Tuzsus[1], Jan Peters[1]

1 Department of Psychology, Biological Psychology, University of Cologne, Cologne, Germany, 2 Biological Psychology of Decision Making, Institute of Experimental Psychology, Heinrich Heine University Duesseldorf, Duesseldorf, Germany

* dmathar@uni-koeln.de

**Data Availability Statement:** All modeling code, behavioral data and physiological data are available on the Open Science Framework, via https://osf.io/

## Abstract

Supplementation with the catecholamine precursor L-Tyrosine might enhance cognitive performance, but overall findings are mixed. Here, we investigate the effect of a single dose of tyrosine (2g) vs. placebo on two catecholamine-dependent trans-diagnostic traits: model-based control during reinforcement learning (2-step task) and temporal discounting, using a double-blind, placebo-controlled, within-subject design (n = 28 healthy male participants). We leveraged drift diffusion models in a hierarchical Bayesian framework to jointly model participants' choices and response times (RTS) in both tasks. Furthermore, comprehensive autonomic monitoring (heart rate, heart rate variability, pupillometry, spontaneous eye blink rate) was performed both pre- and post-supplementation, to explore potential physiological effects of supplementation. Across tasks, tyrosine consistently reduced participants' RTs without deteriorating task-performance. Diffusion modeling linked this effect to attenuated decision-thresholds in both tasks and further revealed increased model-based control (2-step task) and (if anything) attenuated temporal discounting. On the physiological level, participants' pupil dilation was predictive of the individual degree of temporal discounting. Tyrosine supplementation reduced physiological arousal as revealed by increases in pupil dilation variability and reductions in heart rate. Supplementation-related changes in physiological arousal predicted individual changes in temporal discounting. Our findings provide first evidence that tyrosine supplementation might impact psychophysiological parameters, and suggest that modeling approaches based on sequential sampling models can yield novel insights into latent cognitive processes modulated by amino-acid supplementation.

## Author summary

Model-based control during reinforcement learning and temporal discounting have emerged as two candidates delivering on the promise of the Research Domain Criteria as

p78wc/. Notably, raw video recordings from spontaneous eye blink rate assessment are not available due to data privacy reasons. All participants signed a written informed consent prior to participation stating that the respective video files showing their faces will not be shared and that the videos will be deleted after being evaluated by two independent researchers at the end of the year of their participation at the latest. However, eye blink rates from all assessments are available in the form of a text file.

**Funding:** This work was funded by Deutsche Forschungsgemeinschaft (PE1627/5-1 to J.P.) The funders had no role in study design, data collection and analysis, decision to publish, or preparation of the manuscript.

**Competing interests:** The authors have declared that no competing interests exist.

trans-diagnostic characteristics that seem to go awry in a broad range of psychiatric illnesses. Both processes rely on catecholamine transmission and thus may benefit from L-tyrosine supplementation. Here, we tested this in a placebo controlled within-subjects design. On a physiological level, tyrosine intake reduced participants' arousal as revealed by increases in pupil dilation variability and reductions in heart rate compared with placebo. With respect to task performance, we found that following tyrosine intake participants' response times decreased consistently in both tasks without deteriorating task performance. Hierarchical drift diffusion modeling linked this with attenuated decision-thresholds in both tasks. Despite this reduced speed-accuracy trade-off, tyrosine seemed to increase goal-directed control as evident in increased reliance on model-based computations during reinforcement learning and if anything reduced temporal discounting of rewards. Our findings highlight that deeper insights into cognitive effects of supplementation can be gained from applying comprehensive computational models.

## Introduction

The market for dietary supplements that promise to boost cognitive functioning is consistently growing [1]. The freely-accessible catecholamine precursor L-Tyrosine has increasingly gained attention in cognitive neuroscience. Tyrosine is converted to l-3,4-dihydroxyphenylalanine (L-DOPA) by the rate-limiting enzyme tyrosine hydroxylase and then catalyzed into dopamine. Downstream, β-hydroxylase converts dopamine to noradrenaline [2]. As tyrosine hydroxylase is saturated with tyrosine by only approximately 75%, dietary supplementation of tyrosine can increase dopamine and noradrenaline synthesis and transmission [3]. Accordingly, accumulating evidence suggests that tyrosine may boost cognitive functions that rely on catecholamine neurotransmission [4–8]. To date, existing work has mostly focused on working memory performance and perceptual-motor skills in demanding situations that may result in depleted catecholamine levels within the brain [9–12]. Despite these promising findings, there exist no studies yet that looked into potential effects of tyrosine supplementation on other catecholamine dependent cognitive functions that are highly relevant in the context of everyday decision-making.

To foster the identification of core cognitive mechanisms that underlie distinct psychiatric disorders the National Institute of Mental Health has proposed the Research Domain Criteria (RDoC). Two promising candidates that seem to deliver on the RDoC promise are model-based reinforcement learning [13,14] and temporal discounting of rewards [15–17]. Learning from previous experience is essential for optimal behavior in volatile environments and is tightly linked to reward prediction errors signaled by dopaminergic midbrain neurons [18–20]. Reliance on stimulus-reward associations is referred to as model-free reinforcement learning. In contrast, model-based control during reinforcement learning delineates a computationally more demanding incorporation of a cognitive model of the environment to facilitate goal-directed action selection [13,21,22]. The balance between model-free and model-based reinforcement learning has been linked to striatal dopamine transmission [23] and may indeed constitute a trans-diagnostic marker across a range of psychopathologies [14,24–26]. However, reported effects of pharmacological modulations of dopamine transmission on the balance of model-free and model-based reinforcement learning remain mixed [27,28].

Steep temporal discounting constitutes another trans-diagnostic marker with relevance for a variety of psychiatric disorders [15,16,29]. Temporal discounting refers to the trade-off between smaller but sooner (SS) and larger but later (LL) rewards [17]. Typically, both animals

and humans prefer sooner over later rewards. The degree of temporal discounting is highly stable over time within individuals [26,30,31] and has been linked to dopaminergic function within the striatum [32]. Parkinson's disease patients show attenuated temporal discounting when tested on vs. off dopaminergic medication [33]. In line with this finding, de Wit et al. (2002) [34] found that acute administration of D-amphetamine decreased temporal discounting in healthy volunteers. However, a later study did not replicate this effect [35]. With respect to a direct modulatory role of dopamine, findings also appear mixed. Some studies have shown an increase in temporal discounting following moderate increases in dopamine transmission [36], and others showing the opposite [34] or no modulatory effect [37]. Recently, Wagner et al. (2020) [38] revealed attenuated temporal discounting following a single dose (2mg) of haloperidol, a dosage that might increase extra-synaptic striatal dopamine levels via blocking of presynaptic D2 auto-receptors.

Despite the relatively large body of work on the effects of pharmacological modulation of dopamine transmission on reinforcement learning and temporal discounting, no studies to date have reported upon possible modulatory effects of freely available tyrosine supplementation. In addition to counteracting decrements of cognitive performance in working memory tasks [6,9] a few studies indicated that a single dose of tyrosine administration may improve a wider range of cognitive functions, including cognitive flexibility, inhibitory control, working memory, and reasoning [4,5,7,8]. There is also evidence for positive effects of long-term tyrosine intake on cognitive performance, reflected in associations between daily tyrosine intake and working memory, episodic memory, and fluid intelligence [11]. Thus, it seems reasonable that tyrosine supplementation may also modulate model-based reinforcement learning and temporal discounting, two catecholamine dependent trans-diagnostic characteristics. If this is indeed the case, this might have implications for a range of maladaptive behaviors that are implicated in a range of (sub-) clinical conditions [14,15,29].

Here we examined the effect of a single dose of tyrosine (2g) on model-based reinforcement learning and temporal discounting in a double-blind within-subject placebo-controlled study. Importantly, we expanded upon previous work on supplementation effects in two ways. First, we utilized a combination of temporal discounting and reinforcement learning models with drift diffusion model based choice rules [38–42]. This approach yields a more comprehensive account of participants' learning and decision-making behavior by decomposing observed response time (RT) distributions into latent underlying processes. Second, we comprehensively assessed effects of tyrosine supplementation on physiological proxy measures of catecholamine function, specifically spontaneous eye blink rate, pupil dilation, heart rate and variability of the latter two. Physiological measures were obtained both pre and post tyrosine/placebo administration. Results from these exploratory analyses revealed initial evidence for physiological effects of tyrosine supplementation and may inform future supplementation studies by providing potentially objective read-outs regarding te efficacy of supplementation in humans.

## Materials and methods

### Ethics statement

The study was approved by the ethics committee of the German Psychological Association (DGPS; approval number: JP072017). All volunteers provided written informed consent prior to participation.

### Participants

Thirty male healthy volunteers (right-handed, non-smoking, with no history of psychiatric or neurological illness, no medication or drug use), participated in the study. With this, we aimed

at a larger sample-size than previous work regarding effects of tyrosine supplementation on cognitive performance [8,43,44]. We invited only male participants due to the potential impact of hormone levels on central tyrosine levels [45,46]. All volunteers received a reimbursement of 10 € per hour for their participation. Performance dependent additional reimbursement is outlined in the task specific sections further below. Two participants were excluded prior to data analysis, due to clinically significant depressive symptoms (BDI-II > 28), and a diagnosed psychiatric condition in the past.

## General procedure

A double-blind, placebo-controlled, randomized within-subjects design was employed. The study consisted of two testing sessions performed on two separate days with a 7–28 day interval in between. Participants were asked to refrain from drinking alcohol and eating protein-rich food on the evening before each testing day, to reduce possible interactions between tyrosine and other amino acids that might compete for amino acid transporter at the blood-brain barrier [47]. Participants were instructed to fast at least three hours before arriving and refrain from drinking anything except water. Testing started between 9 a.m. and 1 p.m. and lasted approximately 2.5 h.

Each testing day started with a five minute physiological baseline assessment (t0) including spontaneous eye blink rate, heart rate, and pupil dilation. After the baseline physiological recording session, tyrosine/placebo was provided to the volunteers (see Tyrosine/ Placebo administration section below). This was followed by a 60 minutes break in which participants were free to rest and to read in a provided newspaper. Subsequently, participants underwent a second five minute physiological monitoring (t1) for exploratory analysis of possible tyrosine-related effects on physiological arousal. Then (75 minutes following tyrosine/PLC intake) participants performed the sequential reinforcement learning task, followed by the temporal discounting task with a short resting break in between. The order of the tasks was fixed in both sessions. The sequential reinforcement learning task is more challenging and time-consuming, and previous studies found an effect of tyrosine supplementation mainly in or after cognitively demanding situations (e.g. [4]). In total, both tasks lasted for 45 minutes. The first testing day ended with a brief demographic and psychological screening (see below). At the end of the second testing day, participants received the payment and were asked to guess in which session they received tyrosine.

## Physiological data acquisition

For quantification of possible tyrosine-related effects on physiological arousal, we assessed three different measures of autonomic nervous system activity, spontaneous eye blink rate, heart rate, and pupil dilation for five minutes. Spontaneous eye blink rate is discussed as a proxy measure for central dopamine levels [48–50], although more recently this has been called into question in humans [51,52]. Heart rate, pupil dilation and variability at rest in these measures are tightly linked to the interplay of sympathetic and parasympathetic afferents [53–55] and noradrenaline transmission [56–59]. In each of the two physiological assessments per testing day participants were seated in a shielded, dimly lit room 0.6m from a 24-inch LED screen (resolution: 1366 x 768 px; refresh rate: 60 Hz) with their chin and forehead placed in a height-adjustable chinrest. Stimulus presentation was implemented using Psychophysics toolbox (Version 3.0.14) for MATLAB (R2017a; MathWorks, Natick, MA). They were instructed to move as little as possible and fixate a white cross on a grey background presented on the screen. Spontaneous eye blink rate was recorded using a standard HD webcam placed above the middle of the screen. A single recording duration of 5 minutes has been shown to

suffice for assessing stable spontaneous eye blink rate values [60,61]. Pupillometry data were collected using a RED-500 remote eye-tracking system (sampling rate (SR): 500 Hz; Sensomotoric Instruments, Teltow, Germany). heart rate data was acquired utilizing BIOPAC Systems hard- and software (SR: 2000Hz; MP 160; Biopac systems, Inc). For cardiovascular recordings an ECG100C amplifier module with a gain of 2000, normal mode, 35 Hz low pass notch filter and 0.5 Hz/1.0 Hz high pass filter was included in the recording system. Disposable circular contact electrodes were attached according to the lead-II configuration. Isotonic paste (Biopac Gel 100) was used to ensure optimal signal transmission.

## Physiological data analyses

**Spontaneous eye blink rate.**    Spontaneous eye blink rate per minute was quantified by replaying the recorded videos and counting the single blinks manually by two separate evaluators. In case of a discrepancy $>= 2$ blinks this was repeated. The mean of both final evaluations was used.

## Cardiac data

Heart rate data (5 min recordings) were visually screened and manually corrected for major artifacts. We used custom MATLAB code to detect each R peak within the raw data. Specifically, the data was down sampled from 2000 to 200 Hz. Next, we applied the inverse maximum overlap discrete wavelet transform from MATLAB's Wavelet Toolbox to reconstruct the heart rate data based on the 2nd to the 4th scale of the respective wavelet coefficients. The reconstructed data was squared and respective peaks were detected via MATLAB's findpeaks function. The obtained peak locations were then subjected to a custom R script that used the RHRV toolbox for R [62] to compute heart rate and standard deviation of normal to normal beat intervals (SDNN) as a heart rate variability measure.

## Pupil data

We used custom MATLAB code for pupil data analysis (5min recordings). First, NaN values as a result of eye blinks or the like were removed. Next, pupil data were down sampled from 500 Hz to 50 Hz. Outliers within a moving window of five seconds (250 data points) (mean $\pm$ 2 SD) were removed and linearly interpolated. The remaining data was smoothed using a robust weighted least squares local regression ('rloess' function within MATLAB's Curve Fitting toolbox) with a span of 10 data points. Mean pupil dilation size and its coefficient of variation (standard deviation/mean*100; [63]) were then computed and averaged over both hemispheres for each assessment.

## Tyrosine/Placebo administration

Following previous tyrosine administration studies [6,9], participants were administered a dosage of 2 g tyrosine (supplied by The Hut.com Ltd.) or placebo (2 g cellulose powder (Sigma-Aldrich Co.), solved in 200 ml orange juice (Solevita, Lidl). According to Tam et al. (1990) [64], tyrosine administration enhances tyrosine hydroxylation and causes plasma tyrosine levels to peak approximately 60–120 minutes following intake and remain significantly boosted for up to 8h [65]. Consistent with this, the delay between tyrosine/placebo administration and start of the cognitive assessment was exactly 75 minutes. To test whether the participants were blind to the supplementation condition, we asked them to guess their treatment assignment (placebo—tyrosine / tyrosine—placebo) subsequent to completing the second session.

## Sequential reinforcement learning task

Participants performed 300 trials of a modified version (Fig 1A) of the original two-step task by Daw et al. (2011) [13]. Based on suggestions by Kool et al. (2016) [66] we modified the outcome stage by replacing the fluctuating reward probabilities (reward / no reward) with fluctuating reward magnitudes (Gaussian random walks with reflecting boundaries at 0 and 100, and standard deviation of 2.5; Fig 1B).

In short, each trial comprised two successive decision stages. In the 1st stage (S1), participants chose between two options represented by abstract geometrical shapes. Each S1 option probabilistically led to one of two 2nd stage (S2) states that again comprised two choice options represented by abstract geometrical shapes. Which S2 stage state was presented depended probabilistically on the S1 choice according to a fixed common (70% of trials) and rare (30% of trials) transition scheme. The S2 stage choice options each led to a reward outcome. To achieve optimal performance, participants had to learn two aspects of the task. They had to learn the transition structure, that is, which S1 stimulus preferentially (70%) led to which pair of S2 stimuli. Further, they had to infer the fluctuating reward magnitudes associated with each S2 stimulus. We used different but matched task versions for the two testing days (tyrosine/placebo, counterbalanced). Task versions used different S1 and S2 stimuli, and different S2 random walks. However, S2 Gaussian random walks were matched on variance and mean across task versions.

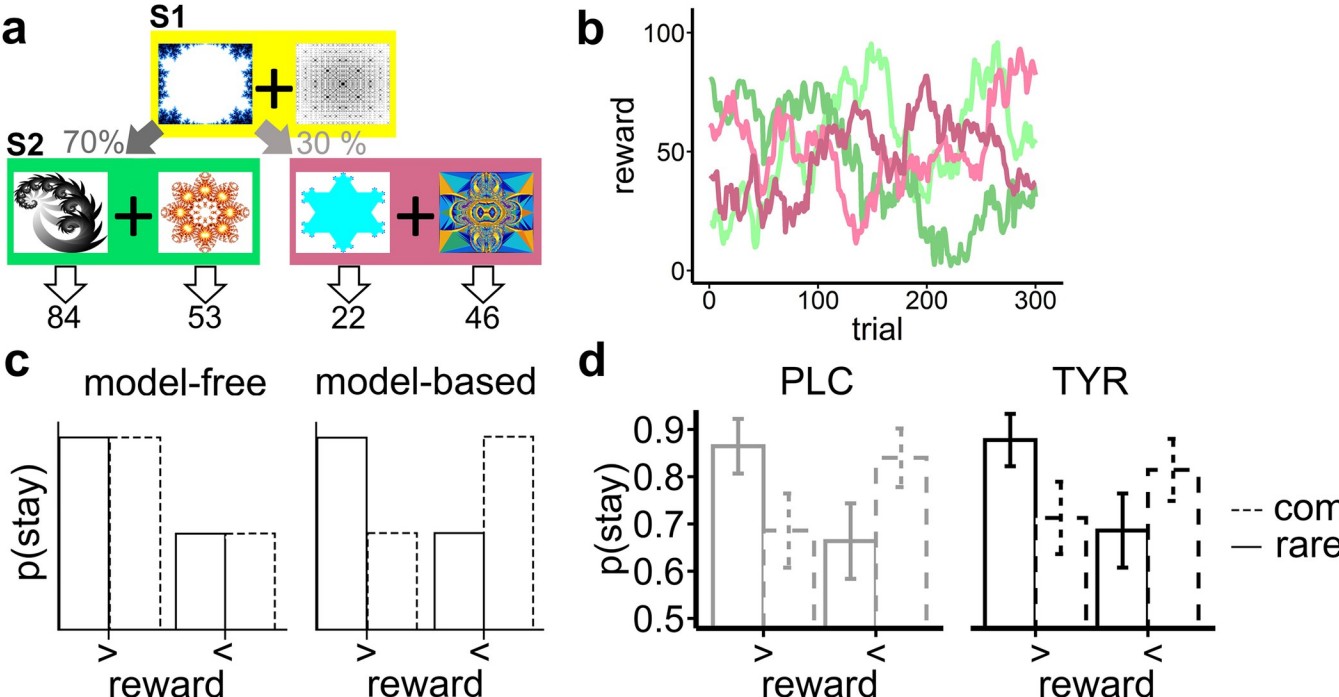

**Fig 1.** (A) Schematic illustration of the seq. reinforcement learning task structure. In each of 300 trials, participants started in stage S1 with a binary choice between two fractals (shown fractals are taken from https://openclipart.org/). Each of the two stimuli was associated with a common (70%) and a rare (30%) transition probability to the two possible states (color coded, here: green / violet) in decision stage S2. In decision stage S2 each stimuli was associated with fluctuating reward points that followed (B) Gaussian random walks with reflecting boundaries at 0 and 100, and standard deviation of 2.5. (C) Putative stay probabilities in decision stage S1 of a pure model-free or pure model-based learner as a function of previous reward height (> / < compared with the preceding 20 trials) and experienced transition probability in the previous trial (common / rare). (D) Participants stay probabilities in stage S1 following placebo (PLC) and tyrosine (TYR).

On each testing day, participants underwent extensive self-paced, computer-based instructions. Instructions provided detailed information about the task structure, the fixed transition probabilities between stages S1 and S2 and the fluctuating reward outcomes in S2. Participants were instructed to earn as much reward points as possible, and that following task completion reward points were converted to € such that they could win a bonus of up to 4€ in addition to their reimbursement of 10€/h. They performed 20 practice trials prior to task performance, with different random walks and stimuli.

### Temporal discounting task

After completing the sequential reinforcement learning task, participants performed 128 trials of a temporal discounting task on each testing day. On each trial, participants selected between a "smaller sooner" (SS) reward available immediately and a "larger later" (LL) reward available after a particular delay in days. SS rewards were fixed to 20 € throughout the trials. LL magnitudes were computed by multiplying the SS reward with [1.01, 1.02, 1.05, 1.10, 1.15, 1.25, 1.35, 1.45, 1.65, 1.85, 2.05, 2.25, 2.65, 3.05, 3.45, 3.85] or [1.01, 1.03, 1.08, 1.12, 1.20, 1.30, 1.40, 1.50, 1.60, 1.80, 2.00, 2.20, 2.60, 3.00, 3.40, 3.8]. The temporal delays of the LL rewards ranged from 1 day to 122 days in eight steps [1, 3, 5, 8, 14, 30, 60,120] days or [2, 4, 6, 9, 15, 32, 58, 119], respectively. LL magnitudes and delays were counterbalanced across the testing days and participants. As in previous studies [38,67] all choice options were hypothetical. Notably, discount rates for real and hypothetical rewards show a high correlation and similar neural underpinnings [68].

Both task were implemented using the Psychophysics Toolbox Version 3 (PTB-3) running under MATLAB (MathWorks).

### Demographic and psychological screening

At the end of the first testing day, participants completed a brief survey on demographic data, socioeconomic status, weight and height (for calculating Body Mass Index (BMI)), trait impulsivity (Barratt-Impulsiveness Scale [69,70]), Behavioral Inhibition and Behavioral Activation System (BIS/BAS scale; German Version: [71,72]), and severity of depressive symptoms (Beck Depression Inventory-II (BDI-II; German Version: [73,74])). Since depressive symptoms can be linked to altered dopamine transmission [75], one participant was excluded from data analysis (BDI-II score > 28). Another participant was excluded due to a diagnosed psychiatric condition in the past.

### Model-agnostic analysis

For the sequential reinforcement learning task we used (generalized) mixed effects regression models to examine S1 stay/shift choice patterns, as well as S1 and S2 RTs. These response variables were modeled as a function of previous reward, state transition (common vs. rare) and supplementation (tyrosine vs. placebo). As a model-agnostic measure of temporal discounting, we examined arcsine-square-root transformed proportions of LL choices as a function supplementation condition (tyrosine vs. placebo) with participants as a random effect using a generalized mixed effects regression approach. In line with our modeling analyses (see below), data were filtered such that implausibly fast RTs (see details on DDM modeling) were excluded.

### Computational modeling and data analysis

**Sequential reinforcement learning task.**   We implemented a slightly modified version of the original hybrid reinforcement learning model for this seq. reinforcement learning task

[13,76] to analyze model-free and model-based contributions to behavior. The model updates model-free state-action values ($Q_{MF}$-values, Eqs 1 and 2) in both stages S1 & S2 via a prediction error scheme (Eqs 3 and 4). In S1, model-based state-action values ($Q_{MB}$) are then computed from the transition and reward estimates using the Bellman Equation (Eq 5). To account for potential modulatory effects of tyrosine vs. placebo supplementation, we included additive 'shift' parameters $s_x$ for each parameter $x$ that were multiplied by dummy-coded supplementation predictors $I_t$ (= 1, $TYR$; = 0, $PLC$).

$$Q_{mf_{s1}}(a_{j,t}) = Q_{mf_{s1}}(a_{j,t}) + \Phi(\eta_1 + s_{\eta 1} * I_t)\delta_{s1,t} + \Phi(\eta_2 + s_{\eta 2} * I_t)\delta_{S2,t} \tag{1}$$

$$Q_{mf_{s2}}(s_{2i,t}, a_{j,t}) = Q_{mf_{s2}}(s_{2i,t}, a_{j,t}) + \Phi(\eta_2 + s_{\eta 2} * I_t)\delta_{S2,t} \tag{2}$$

$$\delta_{S1,t} = Q_{mf_{s2}}(s_{2i,t}, a_{j,t}) - Q_{mf_{s1}}(a_{j,t-1}) \tag{3}$$

$$\delta_{S2,t} = r_{2t} - Q_{mf,S2}(s_{2i,t-1}, a_{j,t-1}) \tag{4}$$

$$Q_{MB}(a_{j,t}) = P(s_{21}|s_1, a_j) \max_{a \in \{a_1, a_2\}} (Q_{mf_{s2}}(s_{21}, a) + P(s_{22}|s_1, a_j) \max_{a \in \{a_1, a_2\}} (Q_{mf_{s2}}(s_{22}, a)) \tag{5}$$

Here, $i$ indexes the two different second stage stages ($S_{21}$, $S_{22}$), $j$ indexes actions $a$ ($a_1$, $a_2$) and $t$ indexes the trials. Further, $\eta_1$ and $\eta_2$ denote the learning rates for S1 and S2, respectively. $\Phi$ is the cumulative density function of the normal distribution that maps learning rates $\eta_1$ and $\eta_2$ (and decay-rate $\gamma$) from [–4,4] onto (0,1). This transformation improves model convergence and the efficiency of the MCMC sampling procedure. S2 model-free $Q$-values are updated by means of reward ($r_{2,t}$) prediction errors ($\delta_{S2,t}$) (Eqs 2 and 4). To model S1 model-free $Q$-values we allow for reward prediction errors at the 2nd-stage (Eq 4) to influence 1st-stage $Q$-values (Eq 1). In addition, $Q$-values of all unchosen stimuli were assumed to decay with decay-rate $\gamma_s$ [77,78] towards the mean of the reward outcomes (0.5) according to Eqs 6 and 7:

$$Q_{unchosen}(s_{k,t}, a_{j,t}) = Q_{unchosen}(s_{k,t-1}, a_{j,t-1}) * \Phi(\gamma_s) + (1 - \Phi(\gamma_s)) * 0.5 \tag{6}$$

$$\gamma_s = \gamma + s_\gamma * I_t \tag{7}$$

with $k \in \{1, 21, 22\}$ indexing the first (S1) and the two second stage (S21, S22) stages.

This learning model was then combined with two different choice rules: softmax action selection, and the drift diffusion model [42].

## Sequential reinforcement learning task, softmax action selection

We first implemented a standard softmax action selection scheme to link learned Q-values with participants' choices. Softmax action selection models choice probabilities as a sigmoid function of value differences [79]. In this regard, S1 choice probabilities are modelled via weighting of S1 model-free and model-based $Q$-values through a softmax function [76]. Note that the formulation used on the present study consists of separate model-based and model-free weights. This formulation is algebraically equivalent to the standard formulation proposed by Daw et al. [13] which uses a single parameter to model the relative contributions of model-based and model-free control [76]. Similarly, S2 stage action selection is modelled as a function of weighted model-free $Q$-values (Eqs 8 and 9). An additional parameter $\rho$ was included to model 1st-stage choice perseveration, $rep(a)$ that is set to 1 if the previous S1 choice was the

same and is zero otherwise.

$$p(a_{j,t} = a|s_{1,t}) = \frac{\exp(\beta_{mb_s} * Q_{mb}(a) + \beta_{mf_s} * Q_{mf_{s1}}(a) + \rho_s * rep(a))}{\sum_{a'} exp(\beta_{mb_s} * Q_{mb}(a') + \beta_{mf_s} * Q_{mf_{s1}}(a') + \rho_s * rep(a'))} \tag{8}$$

$$p(a_{j,t} = a|s_{2,t}) = \frac{\exp(\beta_{2_s} * (Q_{mf_{s2}}(a)))}{\sum_{a'} \exp(\beta_{2_s} * (Q_{mf_{s2}}(a')))} \tag{9}$$

with:

$$\beta_{mb} = \beta_{mb} + s_{\beta_{mb}} * I_t$$

$$\beta_{mf_s} = \beta_{mf} + s_{\beta_{mf}} * I_t$$

$$\rho_s = \rho + s_\rho * I_t$$

$$\beta_{2_s} = \beta_2 + s_{\beta_2} * I_t$$

To account for potential modulatory effects of tyrosine vs. placebo supplementation, we included additive 'shift' parameters $s_x$ for each parameter $x$ that were multiplied by dummy-coded supplementation predictors $I_t$ (= 1, *TYR*; = 0, *PLC*). Note, that this softmax model was used for comparison purposes (*S1 Fig*) for a more advanced drift diffusion model framework delineated below.

## Sequential reinforcement learning task, drift diffusion model (DDM) implementation

We replaced the standard softmax action selection with a series of drift diffusion model (DDM)-based choice rules to more comprehensively examine modulatory effects of tyrosine (vs. placebo) supplementation on choice dynamics. The DDM belongs to the family of sequential sampling models. In these models, binary decisions are assumed to arise from a noisy evidence accumulation process that terminates as soon as the evidence exceeds one of two response boundaries. For each stage S1/S2 of the seq. reinforcement learning task, the upper boundary was defined as selection of one stimulus, whereas the lower boundary was defined as selection of the alternative stimulus. RTs for choices of the alternative option were multiplied by -1 prior to model fitting. Prior to modeling, we filtered the choice data using percentile-based RT cut-offs, such that on a group-level the fastest and slowest one percent of all trials according to RTs were excluded from modeling, and on an individual subject level the fastest and slowest 2.5 percent were further discarded. With this we avoid that outlier trials with implausible short or long RTs bias the results. We then first examined a null model (DDM$_0$) without any value modulation. Trial-wise RTs on each stage S1 & S2 are assumed to be distributed according to the Wiener-First-Passage-Time (*wfpt*):

$$RT_t \sim wfpt(\alpha_i + s_{\alpha_i} * I_t, \tau_i + s_{\tau_i} * I_t, z, \nu_i + s_{\nu_i} * I_t), i = 1, 2 \tag{10}$$

Here, boundary separation parameters $\alpha_i$ model the amount of evidence required before committing to a decision in each stage Si, non-decision time $\tau_i$ model components of the RT that are not directly implicated in the choice process, such as motor preparation and stimulus processing. The starting point bias $z$ models a bias towards one of the response boundaries before the evidence accumulation process starts. This was set to .5 for both stages, as the

boundaries were randomly associated with the choice options on each stage. The drift-rate parameters $v_i$ model the speed of evidence accumulation. Note that for each parameter $x$, we also included a parameter $s_x$ that models potential modulatory effects of tyrosine (vs. placebo) supplementation (coded via the dummy-coded condition predictor $I_t$).

As in previous work [39–41], we then set up hybrid reinforcement learning DDMs with modulation of drift-rates by value differences between the respective choice options, separately for each stage. First, we set up a model with a linear modulation of drift-rates (DDM$_{lin}$) [40]. For S1, this yields

$$v^{lin}_{S1,t} = \beta_{mb} * (Q_{mb[2]} - Q_{mb[1]}) + \beta_{mf_s} * (Q_{mf_{[2]}} - Q_{mf_{[1]}}) + p_s * rep(a') \tag{11}$$

and the drift-rate in S2 is calculated as

$$v^{lin}_{S2,t} = \beta_{S2_s} * (Q_{mf\,S2[2]} - Q_{mf\,S2[1]}) \tag{12}$$

with

$$\beta_{MB_s} = \beta_{mb} + s_{mb} * I_t$$

$$\beta_{MF_s} = \beta_{mf} + s_{mf} * I_t$$

$$\beta_{2_s} = \beta_2 + s_2 * I_t.$$

We next set up a DDM with non-linear (sigmoid) drift-rate modulation (DDM$_S$) that has recently been shown to better account for the value-dependency of RTs compared with the DDM$_{lin}$ [38,39,41,80,81]. In this model, the scaled value difference from Eqs 11 & 12 are additionally modulated by a sigmoid function with asymptote $v_{max_{Si\,s}}$

$$v^s_{Si,t} = \frac{2 * v_{max_{i\,s}}}{1 + exp(-v^{lin}_{Si,t})} - v_{max_{i\,s}} \tag{13}$$

with

$$v_{max_{i\,s}} = v_{max_i} + s_{v_{max_i}} * I_t, \text{and } i = 1, 2.$$

## Temporal discounting task

We applied a single-parameter hyperbolic discounting model to describe how subjective value changes as a function of LL reward magnitude and delay:

$$SV(LL_t) = \frac{A_t}{1 + \exp(k + s_k * I_t) * D_t} \tag{14}$$

Here, $A_t$ is the reward magnitude of the LL option on trial $t$, $D_t$ is the LL delay in days on trial $t$ and $I_t$ denotes the dummy-coded predictor of the supplementation condition. The model has two free parameters: $k$ is the hyperbolic discounting rate (here modeled in log-space) and $s_k$ models a potential additive effect of tyrosine (vs. placebo) on temporal discounting.

## Temporal discounting task, softmax action selection

In a first modeling scheme, similar to the modeling of the seq. reinforcement learning data, we used a softmax function to link subjective values of LL and SS rewards in each trial with

participants' choices:

$$P(LL)_t = \frac{\exp((\beta + s_\beta * I_t) * SV(LL_t))}{\exp((\beta + s_\beta * I_t) * SV(SS_t)) + \exp((\beta + s_\beta * I_t) * SV(LL_t))} \tag{15}$$

Here, $SV$ is the subjective value of the larger but later reward according to Eq 14 and $\beta$ is the inverse temperature parameter, modeling choice stochasticity. $SV(SS_t)$ was fixed at 20 and $I_t$ is again the dummy-coded predictor of supplementation condition, and $s_\beta$ models a potential modulatory effect of tyrosine on $\beta$. Similar to the modeling framework of the reinforcement learning task, this softmax model was used as a reference model for comparison purposes (S2 Fig) for a drift diffusion model implementation delineated below.

## Temporal discounting task, drift diffusion model (DDM) implementation

In a next step, we replaced softmax action selection with a series of drift diffusion model (DDM)-based choice rules. In all DDM implementations, the upper boundary was defined as the selection of the LL option, whereas the lower boundary was defined as choosing the SS option. RTs for choices of the SS option were multiplied by -1 prior to model fitting. We used a percentile-based cut-off similar to the one described in the seq. reinforcement learning modeling section. We again first implemented a null model (DDM$_0$) without any value modulation of the drift-rate $v$:

$$RT_t \sim wfpt(\alpha + s_\alpha * I_t, \tau + s_\tau * I_t, z + s_z * I_t, v + s_v * I_t) \tag{16}$$

In contrast to the reinforcement learning model, here the starting point bias $z$ was fitted to the data, such that z>.5 reflected a bias towards the LL boundary, and z < .5 reflected a bias towards the SS boundary. As in previous work [39–41], we then set up temporal discounting DDMs with a modulation of drift-rates by the difference in subjective values between the LL and SS options. First, we set up an implementation with a linear modulation of drift-rates (DDM$_{lin}$) [40]:

$$v_t^{lin} = (v_{coeff} + s_{v_{coeff}} * I_t) * (SV(LL_t) - SV(SS_t)) \tag{17}$$

We next examined a DDM with non-linear (sigmoid) trial-wise drift-rate scaling (DDM$_S$) that has recently been reported to account for the value-dependency of RTs better than the DDM$_{lin}$ [38,39,41]. In this model, the scaled value difference from Eq 17 is additionally passed through a sigmoid function with asymptote v$_{max}$:

$$v_t^s = \frac{2 * v_{max_s}}{1 + \exp(-v_t^{lin})} - v_{max_s}, \tag{18}$$

with

$$v_{max_s} = v_{max_s} + s_{v_{max_s}} * I_t \tag{19}$$

All parameters were again allowed to vary according to the supplementation condition, such that we included $s_x$ parameters for each parameter $x$ that were multiplied with the dummy-coded condition predictor $I_t$.

## Hierarchical bayesian model estimation

Models were fit to all trials (after exclusion of implausibly short or long RTs) from all participants using a hierarchical Bayesian modeling approach with separate group-level distributions

for all parameters of the placebo (baseline) condition and additional shift parameters $s_x$ to model tyrosine specific effects on all parameters. Model fitting was performed using MCMC sampling as implemented in STAN [82] running under R (Version 3.5.1) and the rSTAN package (Version 2.21.0). For baseline group-level means, we used uniform and normal priors defined over numerically plausible parameter ranges (see code and data availability section for details). For all $s_x$ parameters modeling tyrosine-related effects on model parameters, we used normal priors with means of 0 and numerically plausible standard deviations (range = 1–10). For group-level standard deviations we used half-cauchy distributed priors with location = 0 and scale = 2.5 for the temporal discounting data and uniform priors within pausible ranges (mean = 0, range standard deviation = 10–50). Sampling was performed with four chains, each chain running for 4000 (6000 for the seq. reinforcement learning data) iterations without thinning after a warmup period of 3000 (5000 for the seq. reinforcement learning data) iterations. Chain convergence was then assessed via the Gelman-Rubinstein convergence diagnostic $\hat{R}$ with $1 \leq \hat{R} \leq 1.02$ as acceptable values. For both tasks, relative model comparison was performed via the *loo*-package in R (Version 2.4.1) using the Widely-Applicable Information Criterion (WAIC) and the estimated log pointwise predictive density (elpd) which estimates the leave-one-out cross-validation predictive accuracy of the model [83]. We then report posterior group distributions for all parameters of interest as well as their 80% and 90% highest density intervals (HDI). For tyrosine-related effects, we report Bayes Factors for directional effects of parameter distributions of $s_x$, estimated via kernel density estimation using R via the RStudio (Version 1.3) interface. These are computed as the ratio of the integral of the posterior difference distribution from 0 to $+\infty$ vs. the integral from 0 to $-\infty$. Using common criteria [84], we considered Bayes Factors between 1 and 3 as anecdotal evidence, Bayes Factors above 3 as moderate evidence and Bayes Factors above 10 as strong evidence. Bayes Factors above 30 and 100 were considered as very strong and extreme evidence respectively, whereas the inverse of these reflect evidence in favor of the opposite hypothesis.

## Posterior predictive checks

We carried out posterior predictive checks to examine whether models could reproduce key patterns in the data, in particular the value-dependency of RTs [41] and of participant's choices. For the seq. reinforcement learning task, we extracted 500 unique combinations of posterior parameter estimates from the respective posterior distributions and used these to simulate 500 datasets using the Rwiener package (Version 1.3.3). We then show median RTs of observed data and the median RTs from the 500 simulated datasets for all DDMs as a function of value differences. Value differences for S1 were computed as the absolute difference between the maximum $Q_{mf_{s2}}$ values of each S2 stage weighted by their respective transition probability. Value differences in S2 were computed as the difference in the actual reward values of the respective choice options. Similarly, we show that our models capture the dependency of participants' stay probabilities in S1 on S1 value differences, and the dependency of their fraction of optimal (max[reward]) choices in S2 on S2 reward differences. For the intertemporal choice task, we binned trials of each individual participant into five bins, according to the absolute difference in subjective LL vs. SS ("decision conflict"), computed according to each participant's median posterior $k$ parameter from the DDM$_S$ separately for the placebo and tyrosine condition. For each participant and condition, we then plotted the mean observed RTs and the percentage of LL choices as a function of decision conflict, as well as the mean RTs and fraction of LL choices across 500 data sets simulated from the posterior distributions of the DDM$_0$, DDM$_{lin}$ and DDM$_S$ and softmax model (choices only).

## Results

### Blindness to the supplementation regime

We assessed whether participants were able to guess on which testing day they received tyrosine. Notably, with 13 (46%) participants having guessed their supplementation regime correctly we found that this was not above chance level ($\chi^2$ = .143, p = .71).

### Physiological data

Participants underwent two physiological arousal assessments per session (five minutes each), assessing spontaneous eye blink rate, pupil dilation and heart rate, as well as variability of the latter two. The first assessment was conducted after participants entered the lab as a baseline (t0). The second assessment was conducted 60 minutes following placebo/tyrosine administration (t1). For each measure, we computed the intra-class correlation coefficient to assess the test-retest reliability between the two baseline sessions, and % change at t1 compared to t0 following placebo and tyrosine to assess potential tyrosine-related modulation of physiological arousal, respectively. As expected, spontaneous eye blink rate showed substantial inter-individual variability (range = 1.58–48.26; Fig 2A), but exhibited good test-retest reliability (Table 1 and Fig 2A). Tyrosine did not significantly modulate spontaneous eye blink rate compared to

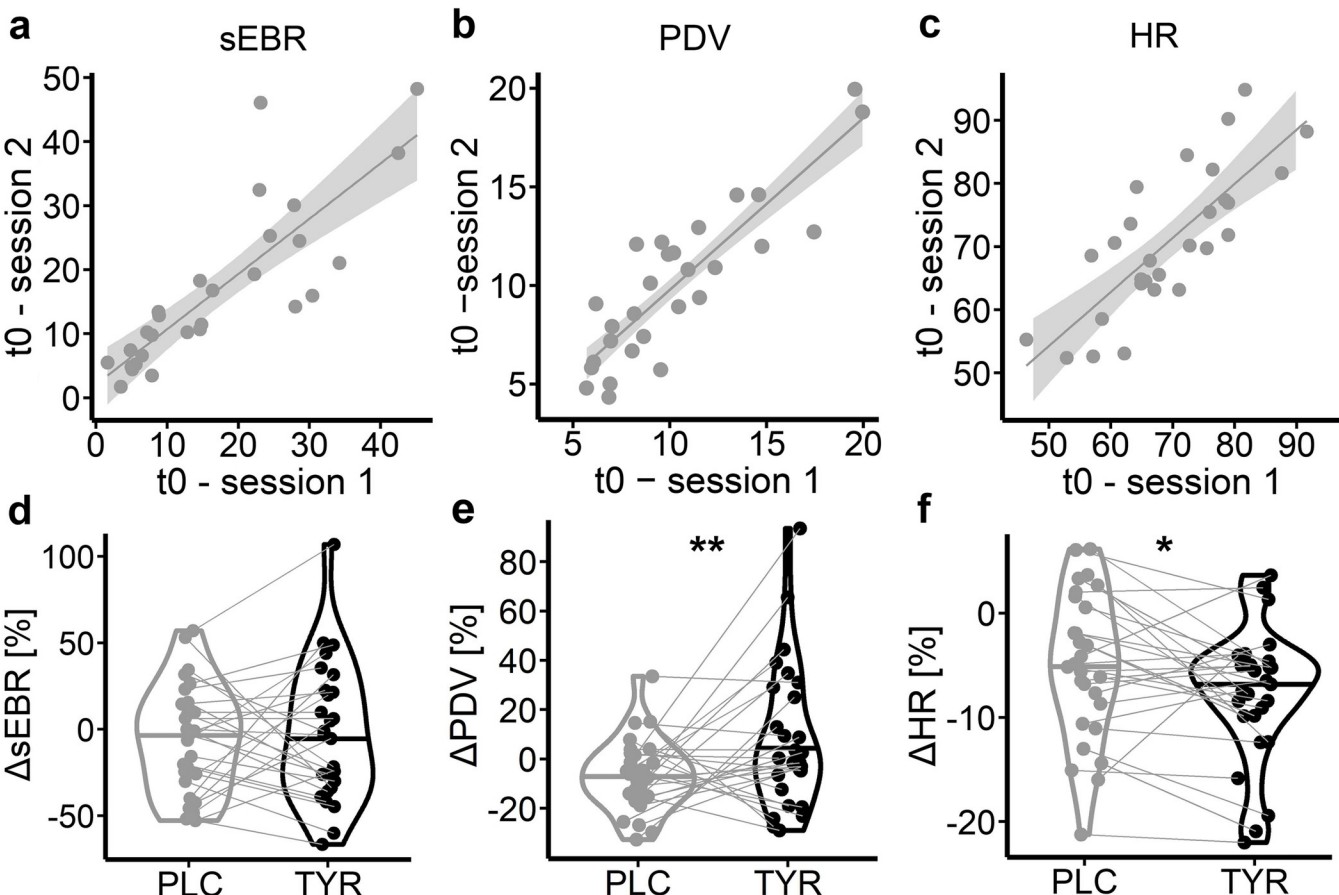

**Fig 2.** A-C: test-retest reliability across the two baseline (t0) sessions for (A) spontaneous eye blink rate (sEBR, blinks per minute), (B) pupil dilation (PD) variation (standard deviation/mean*100) and (C) heart rate (HR). D-F: changes per measure between t0 and 60min post placebo or tyrosine supplementation (t1). * denotes a significant (p < .05) difference between tyrosine and placebo associated changes from t0 to t1. Sample size N = 28 (per session).

**Table 1. Mean (standard error) of physiological arousal markers during the two baseline sessions, test-retest reliability (intraclass correlation coefficient, p-value), and related percent change thereof 1h post placebo / tyrosine (PLC / TYR) intake.** The last column depicts paired t-tests computed on changes following tyrosine vs. placebo. SEBR: spontaneous eye blink rate per minute; PD: mean pupil dilation size; PDV: pupil dilation variability (standard deviation/mean*100); HR: heart rate; HRV: heart rate variability–standard deviation of normal-to-normal heart beat intervals. Sample size: N = 28 (per measurement session).

| | baseline | test-retest icc (p) | ΔPLC [%] | ΔTYR [%] | Δ(TYR-PLC) t (p) |
|---|---|---|---|---|---|
| SEBR | 16.84 (±2.24) | .84 (4.62*10^-9) | -5.32 (±5.87) | -2.92 (±7.41) | .31 (.76) |
| PD | 3.55 (±.06) | .59 (3.41*10^-4) | -1.38 (±1.23) | -.26 (±1.57) | .75 (.46) |
| PDV | 10.21 (±.72) | .89 (4.06*10^-11) | -6.74 (±2.8) | 8.06 (±5.28) | **>3.34 (.002)** |
| HR | 69.98 (±1.96) | .78 (1.87*10^-7) | -5.02 (±1.33) | -7.72 (±1.18) | **-2.36 (.03)** |
| HRV | 62.03 (±3.76) | .74 (1.87*10^-6) | 11.94 (±5.4) | 15.86 (±4.39) | .68 (.5) |

placebo (Table 1 and Fig 2D). Pupil dilation and variability (standard deviation/mean*100) showed moderate and good reliability, respectively (Table 1 and Fig 2B). While pupil dilation size was not significantly affected by tyrosine (Table 1), changes in pupil dilation fluctuation between t1 and t0 significantly differed between tyrosine and placebo (Table 1 and Fig 2E). Heart rate variability also showed moderate to good test-retest reliability across t0 sessions (Table 1 and Fig 2C). Tyrosine compared with placebo was associated with a stronger heart rate deceleration in t1 compared with t0 (Table 1 and Fig 2F). There was no significant tyrosine-related modulation of heart rate variability between t1 and t0 (Table 1). Note that in the light of the exploratory nature of the physiological data analyses, we report uncorrected p-values.

## Sequential reinforcement learning task

**Model-agnostic analysis.** Tyrosine had no significant effect on participants' average pay-out per trial (tyrosine: 61.39 (± .72) vs placebo: 62.25 ± .5 [mean (±SE)]; t(27) = -1.64, p = .11). Typically, model-agnostic analysis of model-free and model-based contributions to choice behavior in this task are based on the relative impact of previous reward and experienced transition on participants' probability to choose the same stimuli in stage S1 as in the previous trial (Fig 1C). We used a generalized mixed effects regression approach to examine participants' stay/shift behavior in S1as a function of previous reward receipt, state transition (common vs. rare) and supplementation (tyrosine vs placebo) allowing for interactions including participants as a random effect. We observed a main effect of reward ($\beta$ = .13, SE = .04, p = 1.09*10^-3; Table 2 and Fig 1D), reflecting an model-free contribution to behavior, and a reward x transition interaction indicating that subjects incorporated a model-based reinforcement learning strategy ($\beta$ = -.54, SE = .07, p = 1.22*10^-14; Table 2 and Fig 1D). We observed neither a significant effect of condition (tyrosine vs. placebo) nor a significant interaction effect with condition on stay probability in S1 (Table 2 and Fig 1D).

In a similar fashion we examined possible effects of tyrosine supplementation on participants' S1 and S2 RTs. We first tested for a general effect of tyrosine supplementation on trial-wise RTs in a mixed effects regression model, including supplementation as a fixed effect and transition and subjects as random effects. Participants responded faster under tyrosine compared to placebo ($\beta$ = -6.61*10^-3, SE = 1.19*10^-3, p = 2.63*10^-8, Fig 3A). Increased S2 RTs following rare transitions is another indicator of model-based control similar to the above reported reward*transition interaction in S1 choice probabilities [42,85]. Thus, next we computed a mixed effects regression with transition, supplementation and their interaction as fixed effects on S2 RTs, and subject as a random effect. As previously reported (42,87), we observed a significant main effect of transition on S2 RTs ($\beta$ = .06, SE = 7.2*10^-3, p = 2.28*10^-9, Fig 3B).

**Table 2. Effects on participants' (N = 28) probability to choose the same option in S1 as in the previous trial from a generalized mixed effects regression (glmer) analysis (rew = reward; trans = state transition; TYR = tyrosine; glmer model: stay(S1) ~ rew*trans*TYR + (rew*trans + 1 | participant)).** And effects of state transition (trans), TYR vs. placebo (TYR) supplementation and their interaction on participants S2 RTs from a lmer analysis (lmer model: RT(S2) ~ trans*TYR + (trans + 1 | participant).

| S1 –stay probability | β | SE | z/t | p |
|---|---|---|---|---|
| **intercept** | 1.4 | .14 | 9.71 | **2.63*10–22** |
| **rew** | .13 | .04 | 3.27 | **1.09*10^-3** |
| trans | -.06 | .03 | -1.71 | .09 |
| TYR | .01 | .02 | .54 | .59 |
| **rew*trans** | -.54 | .07 | -7.71 | **1.22*10^-14** |
| rew*TYR | .03 | .02 | 1.26 | .21 |
| trans*TYR | -.04 | .02 | -1.44 | .15 |
| rew*trans*TYR | .03 | .02 | 1.21 | .23 |
| *S2 –RT* | | | | |
| **intercept** | 0.7 | 0.02 | 45.74 | **3.97^-27** |
| **trans** | .06 | 7.2*10^-3 | 8.76 | **2.28*10^-9** |
| **TYR** | -4.91*10^-3 | 1.83*10^-3 | -2.67 | **7.63*10^-3** |
| **trans*TYR** | 5.08*10^-3 | 1.84*10^-3 | 2.76 | **5.82*10^-3** |

Tyrosine significantly reduced S2 RTs (β = -4.91*10^-3, SE = 1.83*10^-3, p = 7.63*10^-3, Table 2 and Fig 3B). Furthermore, a significant transition*tyrosine interaction (β = 5.08*10^-3.06, SE = 1.84*10^-3, p = 5.82*10^-3, Table 2 and Fig 3B) reflected a more pronounced slowing of RTs following rare vs. common transitions under tyrosine compared to placebo. Taken together, RT analyses suggest a general RT reduction and an increase of model-based reinforcement learning following tyrosine. Notably, in our modified task version and in contrast to the original sequential reinforcement learning task from Daw et al., (2011) [13], increased model-based reinforcement learning usually leads to increased payout [66]. This was reflected in a significant association of participants' mean payoffs per trial and S2 RT differences between rare and common transitions (placebo(tyrosine): r = .63 (.64), p = 3.45*10^-4 (2.47*10^-4); Fig 4). Likewise, the individual beta weights of the reward*transition interaction

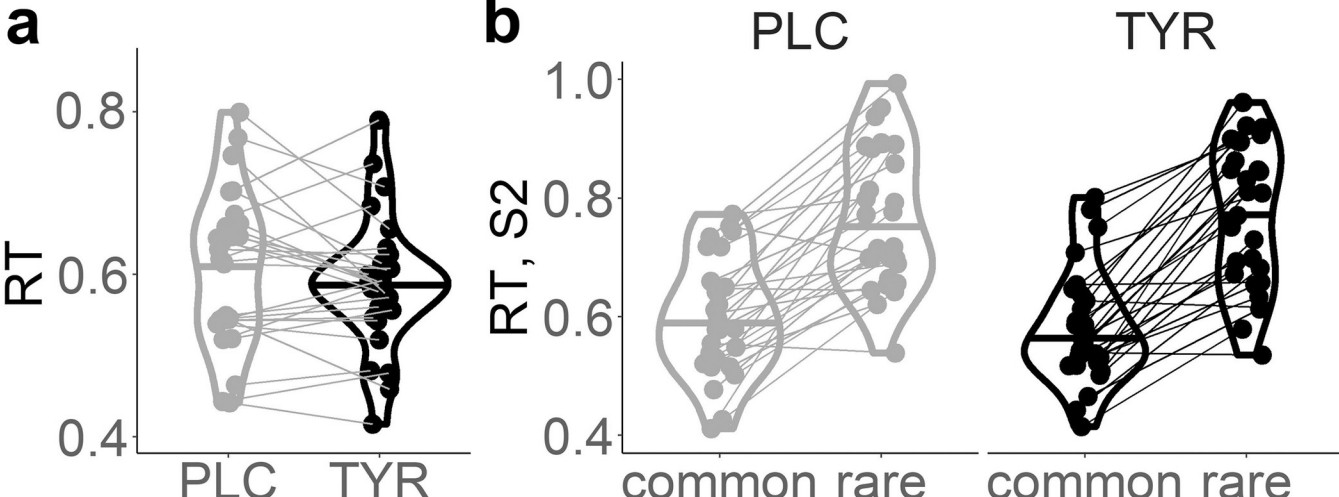

**Fig 3.** Violin plots of (A) median response times (RT) over both stages S1 & S2, and of (B) RTs in S2 subsequent to a common vs. a rare state transition from the sequential reinforcement learning task per supplementation condition (placebo vs. tyrosine).

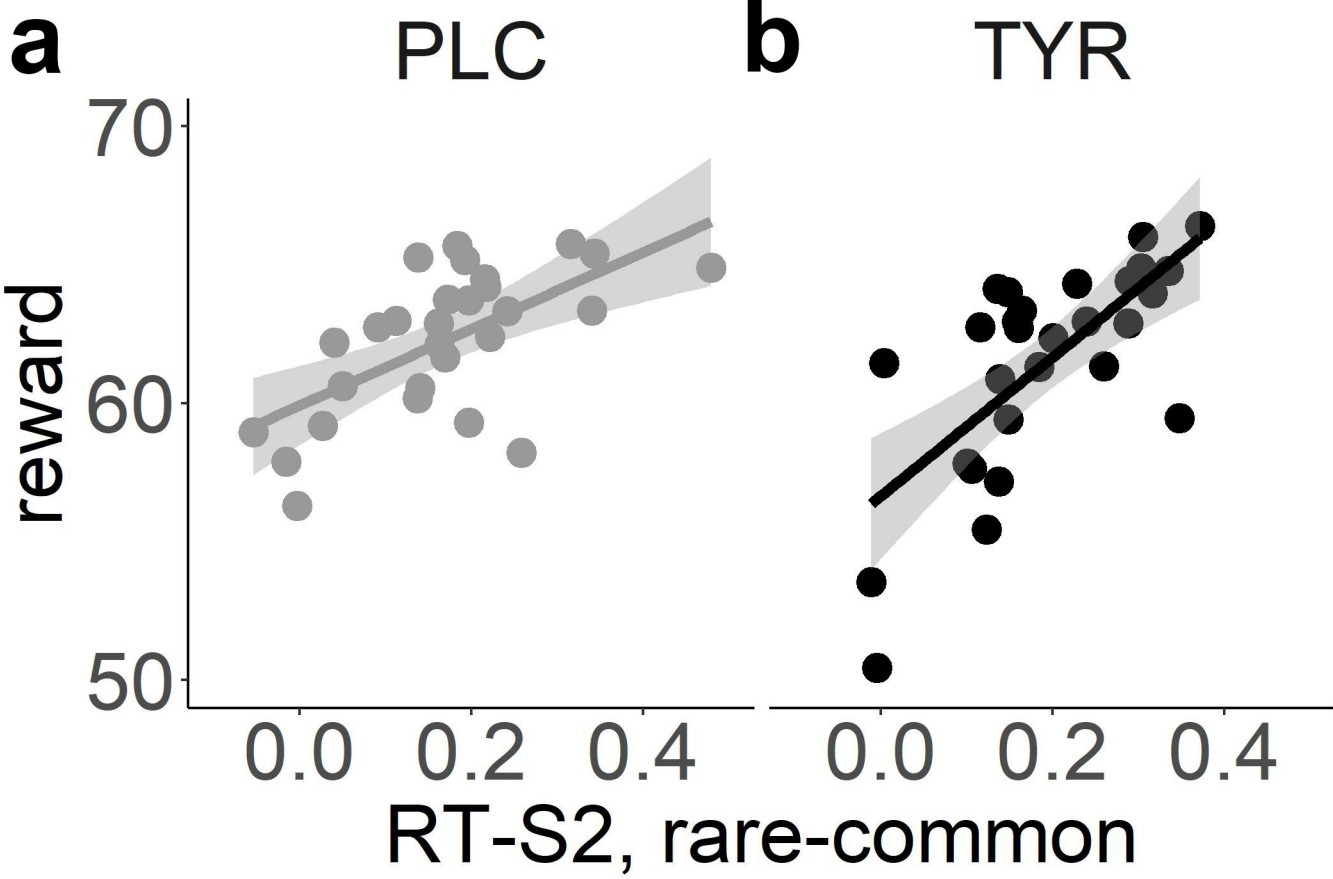

**Fig 4.** Correlation of participants' mean rewards per trial and the differences in S2 RTs following a rare vs. a common state transition for the placebo (A) and tyrosine (B) condition.

effect on participants' stay probabilities reported above showed a highly robust correlation with the mean payoffs (placebo (tyrosine): r = .74 (.72), p = 6.08*10^-6 (1.41*10^-5)).

### Sequential reinforcement learning task, DDM

We used a DDM choice rule to make use of both choices and RT distributions [42,81]. First, we examined the model fit (WAIC, elpd) of three implementations of the DDM that varied in the way they accounted for the modulation of trial-wise S1 & S2 drift-rates by Q-value differences. The DDM with nonlinear drift-rate scaling (DDMs) [38,41,86] accounted for the data best when compared to a DDM with linear scaling (DDM$_{lin}$) [40] and a null model without value modulation (DDM$_0$) (*S3 Table*). Note, that we also compared the three DDMs with a softmax model formulation according to the proportion of correctly predicted binary choices in S1 and S2, respectively. Since the softmax model only accounts for choice data, it shows the highest predictive accuracy. However, the DDM$_s$ predicted participants' choices better than the DDM$_{lin}$ and the DDM$_0$ (S4 Table). Posterior predictive checks for the winning model DDM$_s$ showed that it reproduced the effect decision conflict on participants' RTs and choice patterns (S1 Fig).

We confirmed the expected positive associations between trial-wise drift rates (i.e. $\beta$ weights for model-free-, model-based-, and S2 stage Q-values) and Q-value differences (S5E–S5G Fig, all 90% HDIs > 0). Model-free and model-based $\beta$ weights were also comparable to the

parameters observed in the softmax model (S5I and S5J Fig), indicating robust model-free and model-based contributions during reinforcement learning. Importantly, the individual degree of model-based computations $\beta_{mb}(+s_{\beta_{mb}})$ predicted participants' mean payoff throughout the task (PLC: r = .67, p = 1.06*10^-4; TYR: r = 0.39, p = .04) similar to the observed correlations of model-agnostic measures of model-based control. Notably, tyrosine administration increased model-based computations and choice perseveration (90% HDI > 0; Fig 5F and 5M and Table 3) in stage S1. Likewise, the S1 drift-rate asymptote $v_{max}$, i.e. the maximum rate of evidence accumulation, was reduced following tyrosine compared with placebo (90% HDI > 0; Fig 5H and Table 3). There was also strong evidence for an increase in learning rates in S1 following tyrosine (Fig 5J and Table 3). In line with the observed reduction in RTs following tyrosine, we found attenuated decision-thresholds (90% HDI < 0; Fig 5F and Table 3) and a potential moderate reduction (according to dBF; Table 3) in S2 non-decision times in S2 (Fig 5D and Table 3) under tyrosine. The S2 drift rate coefficient $\beta_2$ was also reduced (Fig 5G and Table 3).

## Temporal discounting task

**Model-agnostic analysis.** We used a generalized mixed effects regression to test for tyrosine-related effects on trial-wise choices in the temporal discounting task with supplementation condition (tyrosine vs. placebo), larger-but-later (LL) reward magnitude (median split), delay (median split) and their interactions as fixed effects and participants as random effect. The likelihood of choosing the LL option was significantly modulated by delay ($\beta$ = -.95, SE = 0.04, p = 1.07*10^-11, Table 4), LL magnitude ($\beta$ = 1.82, SE = .16, p = 3.97*10^-29, Table 4), and their interaction ($\beta$ = -.17, SE = .08, p = .04, Table 4). While we found no main effect of tyrosine on choices (Fig 6A and Table 4), we observed a significant delay*tyrosine interaction ($\beta$ = .09, SE = .04, p = .02, Table 4) reflecting that following tyrosine participants discounted LL rewards less steeply compared to placebo. Similar to the findings in the seq. reinforcement learning task, trial-wise RTs were significantly reduced following tyrosine compared to placebo according to a mixed effects regression model including LL magnitude (median split), delay (median split) and supplementation condition and their interaction as fixed effects and participants as random effects ($\beta$ = -.06, SE = 6.1*10^-3, p = 3.8*10^-23, Table 4 and Fig 6B; see Table 4 for effects of delay & LL magnitude on participants' RTs).

## Temporal discounting task, DDM

As for the the seq. reinforcement learning task data, we first examined the model fit (WAIC, elpd) of three implementations of the DDM that varied in the way they accounted for the modulation of trial-wise drift-rates by value differences. A DDM with nonlinear drift-rate scaling ($DDM_s$) [38,41,86] also accounted for the temporal discounting data best when compared to a DDM with linear scaling ($DDM_{lin}$) [40] and a null model without value modulation ($DDM_0$) (Table 5). We also compared the three DDMs to a softmax model with respect to the proportion of correctly predicted binary choices (SS vs. LL choices). The $DDM_s$ predicted participants' choices numerically on par with the softmax model, whereas the $DDM_{lin}$ and even more so the $DDM_0$ performed substantially worse (S3 Table). Posterior predictive checks of the best-fitting model $DDM_s$ revealed that it accurately reproduced the effect of value differences (i.e. decision conflict) on participants' RTs and the proportion of LL choices (S2 Fig). Note that we previously reported extensive parameter recovery analyses for this model [38,41].

We then examined participants' discounting behavior in greater detail via the posterior distributions of the group-level mean $DDM_s$ parameters (Fig 7). We found no evidence for a general bias towards the SS or the LL option, as the 90% HDI for the starting-point bias parameter

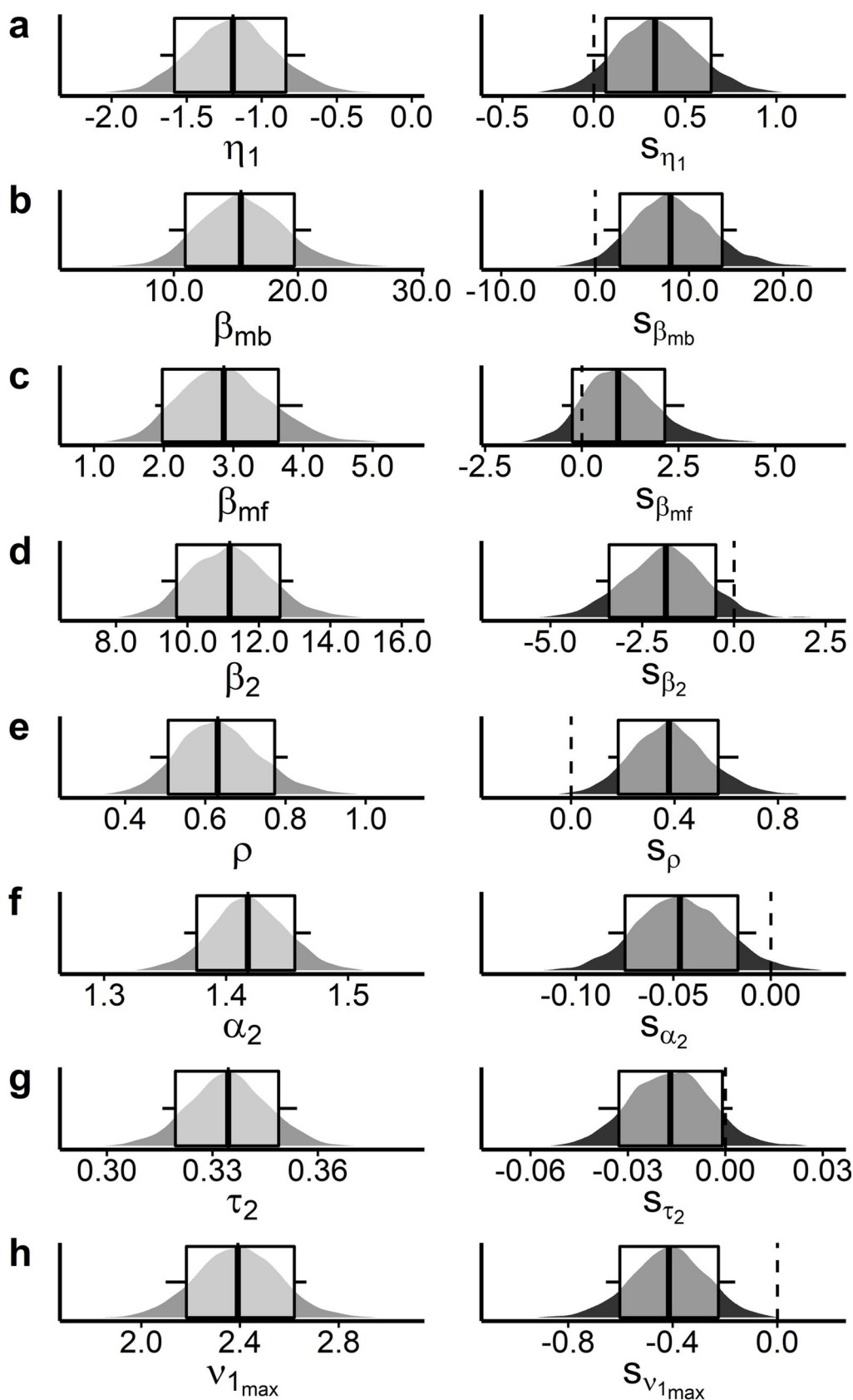

**Fig 5.** Selected posterior parameter distributions of the $DDM_s$ group-level means of the seq. reinforcement learning data following placebo intake (left column, light grey plots) and their respective shifts related to tyrosine intake (right column, dark grey plots). Boxplots depict 80% and 90% HDIs. Depicted parameters are (A) learning rate $\eta_1$ for prediction error related updating in S1, (B) model-based parameter $\beta_{mb}$, (C) model-free parameter $\beta_{mf}$, (D) drift-rate modulation by value differences in S2, (E)(E) perseveration parameter $\rho$, (F) decision threshold in S2 $\alpha_2$, (G) S2 non-decision time $\tau_2$, and (H) S1 drift-rate asymptote $v_{max_1}$. All depicted parameters exhibited strong evidence for a shift following tyrosine except for the model-free parameter $\beta_{mf}$ according to computed directional Bayes Factors (see Table 3). For inspection of all parameter distributions and their related shifts please see S3 Fig).

$z$ overlapped with 0.5 (Fig 7B). Furthermore, trial-wise drift rates increased with increasing value differences, such that the 90% HDI for the drift-rate coefficient parameter $v$ did not include 0 (see Fig 7H).

Tyrosine led to a moderate numerical decrease in temporal discounting according to the $DDM_s$ ($s_k$; Fig 7D and Table 5). Notably, the finding of reduced RTs during temporal discounting following tyrosine were again reflected in substantially reduced decision thresholds ($s_\alpha$; Fig 7F and Table 5) consistent with the similar tyrosine-related findings in the seq. reinforcement learning data. This hints at a general effect of tyrosine supplementation on decision thresholds during value-based decision-making irrespective of the task at hand. Furthermore, tyrosine moderately increased the maximum drift rate ($s_{v_{max}}$; Fig 7L and Table 5).

## Correspondence between task performance and physiology

In exploratory analyses, we tested for associations between task performance (seq. reinforcement learning and temporal discounting) under placebo and physiological arousal measures (spontaneous eye blink rate, pupil dilation, pupil dilation variability, heart rate and heart rate variability), as well as tyrosine-related changes of these. We restricted these exploratory analyses to core aspects of seq. reinforcement learning and temporal discounting, and related evidence accumulation processes to reduce multiple testing burden (see S1 Text for a detailed description). We found that higher pupil dilation at baseline was associated with more impatient (fewer LL) choices during temporal discounting (% LL choices; r = -.55, p = .002; Fig 8A).

**Table 3. Tyrosine related changes in group-level means of the $DDM_s$ for the seq. reinforcement learning data.**
We report mean (and 90% HDIs) for the 'shift' parameters modeling tyrosine-related effects, and Bayes Factors for directional effects (tyrosine>placebo). * (**): Strong evidence (dBF>10 / < .1) for a tyrosine-related effect (& 90% HDIs outside of zero).

|  | mean [90% HDI] | directional BF |
|---|---|---|
| $s_{\alpha_1}$ | -.02 [-.07, .03] | .36 |
| $s_{\alpha_2}$ | **-.05 [-.08, -.01]** | **.03**** |
| $s_{\tau_1}$ | -.01 [-.04, .02] | .38 |
| $s_{\tau_2}$ | **-.02 [-.04, .002]** | **.097*** |
| $s_{\beta_{mf}}$ | 1.01 [-.51, 2.65] | 6.07 |
| $s_{\beta_{mb}}$ | **8.24 [.88, 15.07]** | **37.01**** |
| $s_{\beta_2}$ | **-1.87 [-3.76, .01]** | **0.06*** |
| $s_{v_{max_1}}$ | **-.42 [-.66, -.16]** | **0.003**** |
| $s_{v_{max_2}}$ | .02 [-.43, .44] | 1.09 |
| $s_{\eta_1}$ | **.34 [-.04, .71]** | **14.28*** |
| $s_{\eta_2}$ | 0.2 [-.13, .56] | 4.84 |
| $s_\gamma$ | -.03 [-.31, .27] | .75 |
| $s_\rho$ | **.38 [.14, .65]** | **157.73**** |

**Table 4. Effects on the probability to choose the larger-but-later (LL) option in the temporal discounting task (N = 28) from a generalized mixed effects regression analysis (glmer model: choice(LL)~delay*LL magnitude*tyrosine + (delay*LL magnitude + 1 | participant)).** And effects on participants' RTs in the temporal discounting task following a lmer model (lmer model: rt ~ delay*LL magnitude*tyrosine + (delay*LL magnitude +1 | participant)), with delay & LL magnitude as binary predictors via median split in both mixed effects models.

| LL choices | β | SE | z | p |
|---|---|---|---|---|
| **intercept** | 1.09 | .44 | 2.5 | **.01** |
| **delay** | -.95 | .04 | -6.8 | **1.07*10^-11** |
| **LL magnitude** | 1.82 | .16 | 11.21 | **3.97*10^-29** |
| TYR | .03 | .04 | .66 | .51 |
| **delay*LL magnitude** | -.17 | .08 | -1.98 | **.04** |
| **delay*TYR** | .09 | .04 | 2.29 | **.02** |
| LL magnitude*TYR | -.03 | .04 | -.72 | .47 |
| delay*LL magnitude*TYR | .04 | .04 | 1.18 | .24 |
| *RTs* | | | | |
| **intercept** | 1.23 | .07 | 18.34 | **3.89*10^-17** |
| **delay** | .03 | .01 | 2.46 | **.02** |
| LL magnitude | -.04 | .02 | -1.93 | .06 |
| **TYR** | -.06 | 6.1*10^-3 | -9.95 | **3.8*10^-23** |
| **delay*LL magnitude** | .02 | 8.7*10^-3 | 2.23 | **.03** |
| delay*TYR | -3.19*10^-3 | 6.1*10^-3 | -.52 | .6 |
| LL magnitude*TYR | 7.27*10^-3 | 6.1*10^-3 | 1.19 | .23 |
| delay*LL magnitude *TYR | 4.06*10^-3 | 6.09*10^-3 | .67 | .51 |

In line, higher pupil dilation was associated with a significant bias towards impatient SS choices ($z$; r = -.63, p = 3.0*10^-4; Fig 8B). With respect to tyrosine-related changes in physiological arousal, greater pre-post heart rate changes following tyrosine compared with placebo were associated with tyrosine-related shifts in temporal discounting ($s_{\log (k)}$; r = .61,

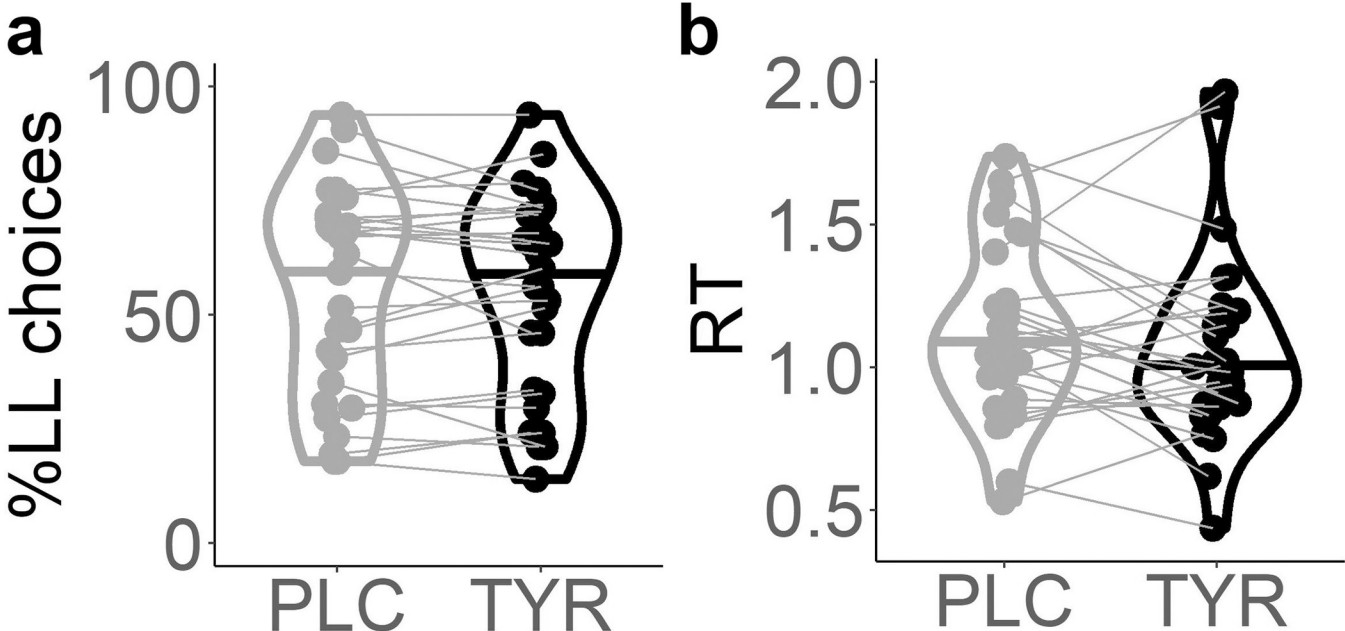

**Fig 6.** Violin plots of (A) proportions of LL choices following placebo and tyrosine supplementation, and of (B) median response times (RT) in the temporal discounting task per supplementation condition (placebo vs. tyrosine).

**Table 5. Tyrosine related changes in group-level means of the DDM$_s$ for the temporal discounting task data.** We report mean and 90% HDIs for the 'shift' parameters and Bayes Factors for directional effects (tyrosine>placebo). * denotes strong evidence (dBF>10 / < .1) for a tyrosine-related effect on the respective shift parameter $s_\alpha$.

|  | mean [90% HDI] | directional BF |
|---|---|---|
| $\log(s_k)$ | -.13 [-.32,.09] | .19 |
| $s_z$ | -.00 [-.02, .02] | 0.91 |
| **$s_\alpha$** | **-.15 [-.31, 0]** | **.06**$^*$ |
| $s_\tau$ | -.01[-.11, .04] | .3 |
| $s_v$ | -0.012 [-.06, .03] | 0.48 |
| $s_{v_{max}}$ | .12 [-.05, .3] | 7.15 |

p = 5.76*10^-4; Fig 8C). All other tested associations were non-significant after adjusting for False Discovery Rate [87]; all p-values > = .005; FDR adjusted p-value = .003).

## Discussion

In a double-blind, placebo-controlled within-subjects design, we investigated the effect of a single dose (2g) of tyrosine supplementation on two trans-diagnostic characteristics of human decision-making, mental health and maladaptive behaviors—sequential reinforcement learning (seq. reinforcement learning) and temporal discounting. We used drift diffusion modeling (DDM) in a hierarchical Bayesian approach to reliably quantify tyrosine-related changes within distinct aspects of the dynamic choice processes in both tasks. Assessment of spontaneous eye blink rate, pupil dilation, heart rate and variability of the latter two both pre and post tyrosine/placebo administration enabled us to associate catecholamine dependent behavior with physiological arousal markers and tyrosine-related modulation of both in an exploratory manner. All physiological measures exhibited moderate to good test-retest reliability. Tyrosine reduced physiological arousal compared to placebo as indexed by modulations of pupil dilation variation and heart rate. On the behavioral level, tyrosine consistently reduced RTs across tasks, without compromising task performance. Hierarchical Bayesian DDMs linked these effects to attenuated decision-thresholds in both tasks. Tyrosine augmented model-based computations during seq. reinforcement learning and, if anything, reduced rather than increased temporal discounting. Participants' mean pupil dilation at baseline predicted core aspects of their temporal discounting behavior. Tyrosine-related reductions of arousal were associated with individual changes in temporal discounting following tyrosine supplementation.

### Tyrosine attenuates physiological arousal

We assessed pupil dilation (mean & variability), heart rate (mean & variability), as well as spontaneous eye blink rate during a five minute interval at baseline (t0) and 60 minutes following tyrosine/placebo supplementation (t1) on both testing days. First, all three measures exhibited moderate to good test-retest reliability across assessments. T0 assessment was conducted only a few minutes after participants entered the lab. Presumably, participants were in state of heightened physiological arousal at this baseline assessment, while an hour later (t1) at the second physiological assessment participants had become accustomed to the lab environment. This was mirrored in a reduced heart rate at t1. Interestingly, the reduction in physiological arousal from t0 to t1 was more pronounced following tyrosine supplementation. This was reflected in a significant difference in the t1-t0 change in pupil dilation and heart rate following tyrosine compared to placebo supplementation. This might appear counterintuitive, as tyrosine is a precursor of dopamine and noradrenaline, and thus tyrosine supplementation

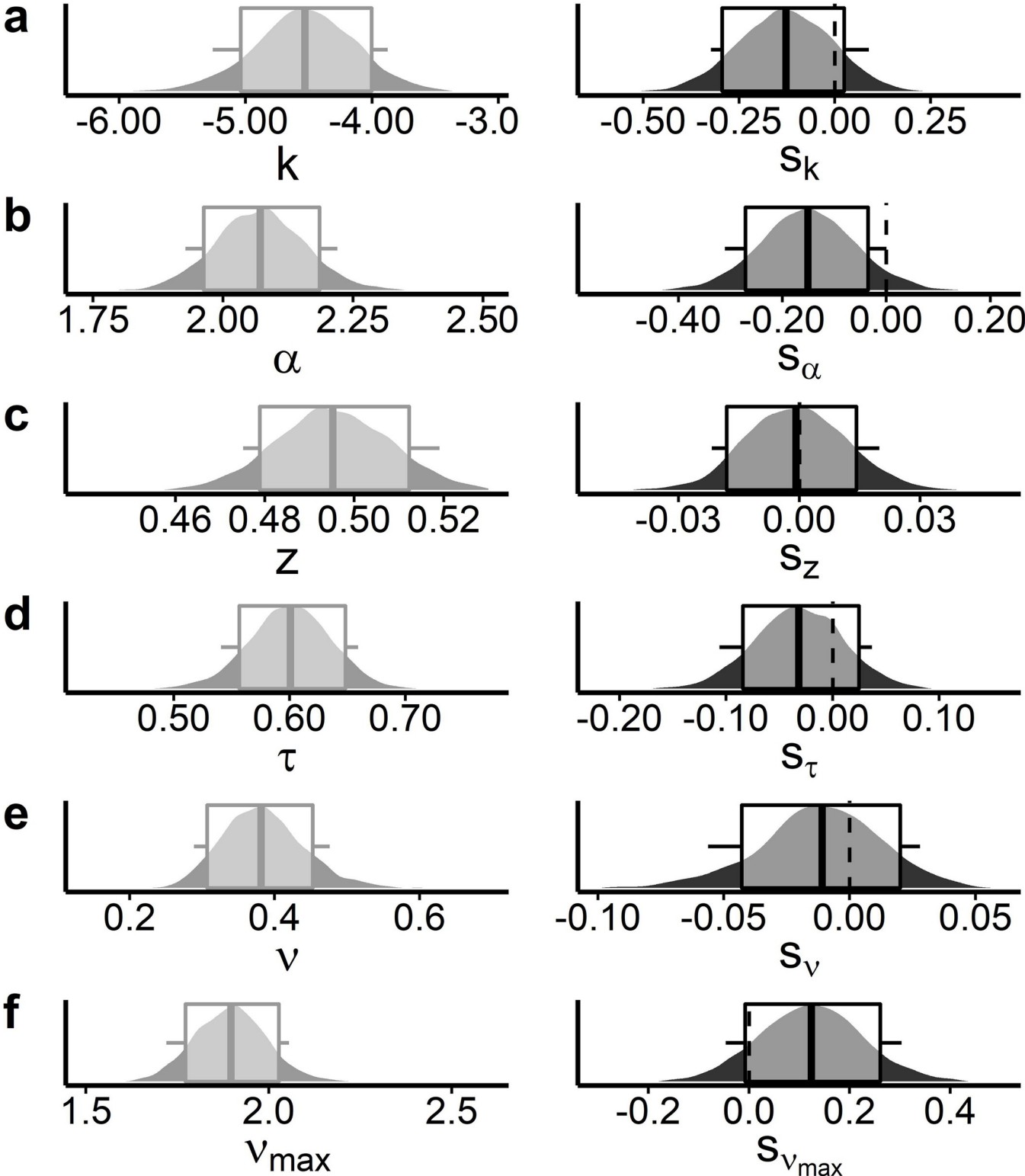

**Fig 7.** Posterior parameter distributions of the DDM$_s$ group-level means of the temporal discounting data following placebo intake (left column, light grey plots) and their respective shifts related to tyrosine intake (right column, dark grey plots). Boxplots depict 80% and 90% HDIs. Depicted parameters are (A) discount rate log($k$), (B) decision-threshold $\alpha$, (C) starting point bias $z$, (D) non-decision time $\tau$, (E)(E) drift-rate $v$ and (F) drift-rate asymptote $v_{max}$; as well as their tyrosine-related ‚shifts'. Only decision-threshold $\alpha$ showed strong evidence for a shift following tyrosine intake according to directional BF and 90% HDI (dBF < .1 and HDI < 0; see Table 5).

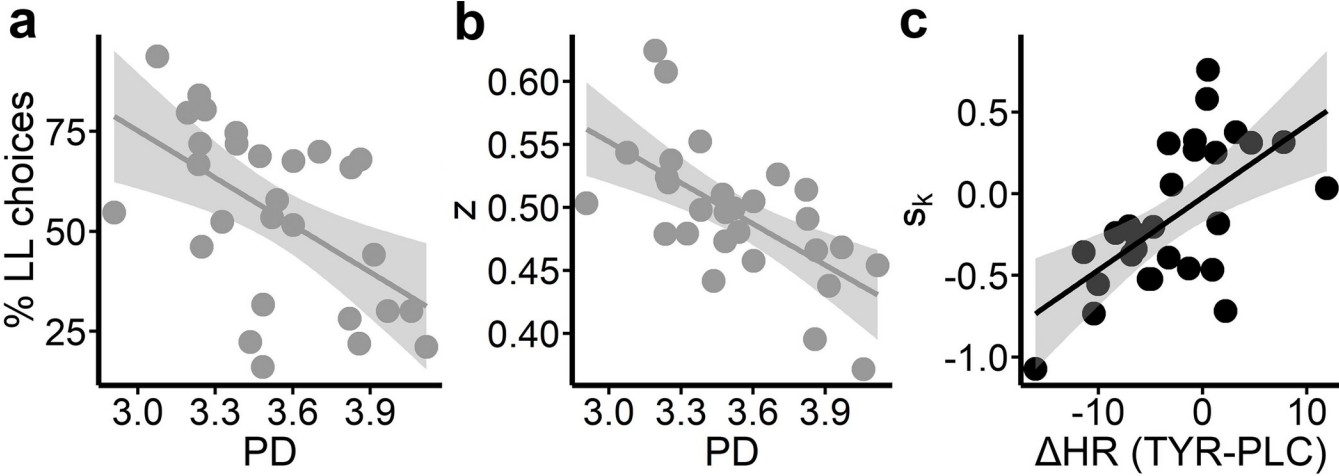

**Fig 8.** Participants (N = 28) pupil dilation (PD) at baseline was predictive of (A) % LL choices, (B) bias towards SS choices ($z$) and (C) tyrosine associated changes in temporal discounting ($s_{\log (k)}$) was linked with supplementation related differences (tyrosine-placebo) in heart rate (HR) changes relative to baseline.

presumably increases dopamine and noradrenaline synthesis capacity and release [65,88,89]. In line with this, increased noradrenaline plasma levels have been linked to tyrosine administration in humans [90]. However, tyrosine supplementation has also been linked to hypotensive effects in humans specifically in stressful situations [10]. Similar effects in rodents are assumed to be related to $\alpha$-adrenergic receptor stimulation through elevated noradrenaline release [91,92]. These findings suggest that, in the present study, tyrosine might have attenuated physiological arousal via noradrenaline-related mechanisms. However, a recent study that tested cognitive and physiological effects of increased noradrenaline transmission reported increased blood pressure as a measure of heightened physiological arousal compared to baseline assessment following administration of the noradrenaline receptor antagonist Yohimbine [93]. Thus, follow-up studies are needed to test whether the observed exploratory effects of tyrosine supplementation on physiological measures can be replicated in a larger sample.

### Tyrosine reduces RTs

Tyrosine supplementation was consistently associated with reduced RTs, both in temporal discounting and seq. reinforcement learning. This complements earlier work from Magill et al. (2003) [7] who found that following sleep deprivation, tyrosine significantly reduced RTs compared with placebo across several cognitive tasks. Likewise, acute tyrosine and phenylalanine depletion significantly slowed RTs during a rapid visual information processing task [94]. Changes in dopamine and/or norepinephrine neurotransmission might underlie these effects. Recently, Wagner et al. (2020) [38] reported reduced non-decision times following administration of the D2 receptor antagonist Haloperidol in healthy volunteers during a temporal discounting task. They interpreted their finding as facilitated motor responding due to increased dopamine transmission via presynaptic action of low dosages of Haloperidol. Findings of a dopaminergic speeding of RTs in tasks testing response vigor [95,96] further support the idea that tyrosine might have reduced RTs via a dopamine-dependent mechanism. However, increased norepinephrine transmission following tyrosine may also have contributed to RT reductions. Pharmacological blockade of central $\alpha_2$-adrenergic receptors was shown to attenuate RTs, but also increase impulsive responding and increase heart rate and blood pressure in healthy human volunteers [97] which however contrasts with our findings regarding temporal

discounting performance and physiological correlates of tyrosine intake. However, Herman et al., (2019) [94] found reduced stop-signal RTs indicating attenuated motor impulsivity and stronger heart rate deceleration compared to baseline following a noradrenergic antagonist in comparison to placebo. These inconsistencies may be attributed to the complexity of the nor-adrenergic system in relation to its cognitive and physiological output depending on the specific receptor family (presynaptic $\alpha_2$ and postsynaptic $\beta$ and $\alpha_1$), the major release site and the balance between tonic and phasic transmission [98].

### Drift Diffusion Modeling (DDM)

We implemented value-based decision (temporal discounting) and reinforcement learning models (model-based reinforcement learning) in a sequential sampling framework (DDM) [26,38–42]. Model comparison replicated previous results [26,38,41,86], such that data in both tasks were consistently better accounted for by models assuming a non-linear trial-wise drift rate modulation. Posterior predictive checks of choices and RTs confirmed the superior repro-duction of relevant inherent patterns (*S1 and S2 Figs*). Prior parameter recovery work con-firmed that DDM group-level parameters for temporal discounting data [38,41] and seq. reinforcement learning data [42] recovered well. Notably, this modeling approach allowed us link the tyrosine-related RT reductions in both tasks to a reduction in decision thresholds (see next section). This highlights the valuable insights that comprehensive modeling approaches can add beyond mere analysis of choices and RTs in neuro-cognitive research.

### Tyrosine reduces decision-thresholds

DDM linked the attenuated RTs following tyrosine supplementation to reduced decision thresholds in both tasks. This parameter models the amount of evidence that is accumulated before committing a decision, i.e. the speed-accuracy trade-off [99,100]. Interestingly, in nei-ther task was this associated with objective performance decrements: Tyrosine did not reduce participants' earnings in the seq. reinforcement learning task nor did it increase impulsive choices during temporal discounting. Conversely, model-based control increased during seq. reinforcement learning as evident in a tyrosine-related increase in S2 RT slowing following rare transitions and increased $\beta_{mb}$ according to the DDM. It is likely that these parallel effects account for the lack of a performance boost under tyrosine. The benefits of improved MB con-trol under tyrosine might simply have been compensated by reduced decision thresholds, which lead to noisier and therefore less optimal choices.

Likewise, if anything, participants showed a moderate decrease in temporal discounting. Decision thresholds have been linked to subthalamic nucleus activity [101]. This part of the hyper-direct cortico-basal ganglia pathway [102] receives direct dopaminergic [103,104] and noradrenergic [103,105] input. Recent human pharmacological work also directly implicates dopamine neurotransmission in the regulation of decision thresholds [106]. A further function of the subthalamic nucleus is the withholding of premature responding to allow for more time to evaluate information and select the most appropriate action [107,108]. Deep brain stimula-tion of the STN in Parkinsonism was shown to attenuate RTs in a reinforcement learning task in high conflict trials while leaving reinforcement learning performance intact [109]. Tyrosine may have modulated STN activity in a similar manner via increased tonic dopaminergic and or noradrenergic input. We also observed consistent numerical reductions in non-decision times across tasks, although strong evidence was restricted to the second stage in the seq. rein-forcement learning task. This may reflect enhanced motor responding e.g. via increased DA transmission [38] within striato-cortical pathways following tyrosine intake. Recently, West-brook et al. (2020) [110] linked striatal dopamine synthesis capacity with motivation to allocate

cognitive effort to obtain rewards. Thus, tyrosine may also have shifted the motivation for task engagement in both tasks that potentially is reflected in shorter RTs and reduced decision thresholds. This explanation can potentially reconcile dopamine theories of response vigor and reward rate maximization with action selection accounts [106].

## Tyrosine impacts model-based computations and perseveration

Tyrosine supplementation was associated with increased model-based computations during reinforcement learning as evident in a tyrosine-related increase in S2 RTs slowing following rare transitions and increased $\beta_{mb}$ according to the DDM. This convergence across model-agnostic and model-based analyses strengthens the confidence in the result. Notably, both measures of model-based control predicted the mean payoffs earned by the participants. Thus, these findings are not mere response time effects, but they are directly linked to task performance. The increase in model-based computations following tyrosine supplementation was not evident in model-agnostic analysis of participants' stay probabilities in decision stage S1 nor in the softmax model. This indicates that participants were more aware of the transition structure following tyrosine supplementation, as evident in the enhanced response time differences for rare vs. common transitions, but they used this knowledge only to marginal extend during their actual choices. DDM in general is more sensitive to these subtle changes in response behavior compared with a one-trial-back based regression analysis and our DDM$_s$ in particular indeed captured participants' choice patterns more precisely than the softmax model as evident in the posterior predictive checks (*S1 Fig*). This highlights the additional insights that response time analysis and DDM can yield regarding supplementation related changes in response patterns. Although we can only speculate about the potential underlying mechanism, this is in line with previous reports on dopamine precursor L-dopa-related increases in model-based control in healthy participants and Parkinson's disease patients [28,111]. But see Kroemer et al. (2019) [27] for contrary findings. Similarly, tyrosine may have augmented dopamine transmission in fronto-striatal circuits, possibly facilitating model-based prediction error encoding [13,23,112]. Tyrosine may also have increased dopaminergic and/or noradrenergic input to the hippocampus, potentially facilitating model-based control [113]. Also, this finding may partly relate to a greater awareness of the transition structure in participants due to enhanced working memory capacity following tyrosine [114]. However, this account will need to be directly tested in future work.

Tyrosine also increased perseveration in S1. We recently reported that L-dopa reduced directed (uncertainty-based) exploration in a restless bandit task [115]. This was mirrored by reduced tracking of overall uncertainty in anterior cingulate and insula cortex. As the present model did not explicitly account for exploration, the observed increases in perseveration might be due to reduced exploration. Further modeling work is required to disentangle perseveration and exploration effects in the seq. reinforcement learning task.

Modeling further revealed a reduced asymptote of the drift-rate in S1 ($v_{max_1}$), an increase in the learning rate of model-free Q-values in S1 ($\eta_1$) and attenuated drift-rate scaling in S2 ($\beta_2$). These findings all suggest a modulation of value updating and related drift rate modulation during reinforcement learning following tyrosine. This resonates with earlier work that has linked dopaminergic transmission to value updating during reinforcement learning [116–118].

## Tyrosine effects on temporal discounting

In contrast to the consistent effects of tyrosine on model-based control, evidence for a reduction in temporal discounting following tyrosine supplementation was somewhat more mixed.

A model agnostic mixed effects regression analysis revealed a significant interaction effect of delay*tyrosine on LL choice proportions, whereas log ($k$) in the DDM was only numerically reduced. While some studies showed similar effects following pharmacological manipulations to foster dopamine transmission [34,38] other studies have reported contrasting effects [36,119]. These inconsistencies may partly be due to potential inverted U-shaped associations between central dopamine levels and cognitive performance [37,120]. However, tyrosine-related changes in temporal discounting were not differentially modulated by blink rate, a potential proxy measure for central dopamine levels, arguing against this account [49,121]. However, a more recent publication from Sescousse et al. (2018) [52] provides a much more critical view on blink rate as a proxy measure for central dopamine levels in humans.

There is also growing evidence for a modulatory role of noradrenaline transmission on measures of impulsivity. In 2006, Chamberlain et al. [122] showed for the first time that increased noradrenaline transmission via the re-uptake inhibitor Atomoxetine improved response inhibition. This finding has been replicated in attention-deficit hyperactivity disorder [123] and Parkinson's disease patients [124] and was also replicated using the noradrenergic antagonist Yohimbine [93]. In contrast, they found no effect on temporal discounting performance. However, they used the Monetary Choice Questionnaire [31] that only incorporates 27 choices and thus may lack sensitivity in comparison to our temporal discounting task with 128 choices that allowed subsequent modeling of choice behavior and response time distributions via hierarchical Bayesian DDM.

## Associations between physiological arousal and temporal discounting

We ran exploratory analyses to link physiological arousal to task performance. Greater mean pupil dilation at baseline (t0) predicted increased temporal discounting across model-agnostic (larger-later choice proportions) and model-based measures (log(k), bias parameter z). This extends previous findings on pupillometry correlates of temporal discounting at the trial level [125,126]. For example, Lempert et al., (2016) [126] found that greater pupil dilation during processing of choice options was associated with an increased probability to select the larger-later reward. In contrast, in the present study, greater pupil dilation *before* task performance predicted *more* temporal discounting. Trial-wise pupillometry measures, reflecting transient changes in arousal thus might show reverse effects compared to trait-like arousal levels at rest. Schmidt et al. (2013) [127] found a negative association of arousal (heart rate) at rest and risk-taking, contrasting with our finding of a positive association of arousal at rest (pupil dilation) and impulsive choice. Thus, future studies of catecholamine dependent cognitive functioning and related supplementation interventions should incorporate state of the art modeling approaches and physiological correlates of arousal to shed light on the mixed findings so far [128].

## Practical relevance and clinical implications

The findings of improved model-based computations and slightly reduced temporal discounting following a single dose of tyrosine supplementation, if replicated in a larger cohort, may be of importance for a range of maladaptive behaviors that are implicated in a spectrum of (sub-)clinical conditions [14,15,29]: In the light of the findings from Kühn et al. (2019) [73] a tyrosine rich diet may prove beneficial not only for improving cognitive functions that rely on catecholamine transmission such as working memory but also for reducing the risk of developing compulsive and addiction-like behavioral patterns. Similarly, individuals suffering from related (sub-)clinical conditions may show an improvement of symptoms via tyrosine supplementation related modulation of these trans-diagnostic cognitive processes. In addition,

heightened tyrosine intake may also increase the motivation to allocate cognitive effort to focus on long-term (therapeutic) goals via enhancement of striatal dopamine synthesis capacity [110,129].

## Limitations

Our study has a number of limitations that need to be acknowledged. First, we only included male participants. This was done in order to minimize inter-individual variability in estrogen levels known to modulate central DA transmission with potential implications for tyrosine effects [45,46]. Second, our final sample size of n = 28 is relatively small. However, when compared to earlier supplementation work it still is among the largest within-subject studies on tyrosine supplementation reported so far [9]. Future studies would nonetheless benefit from larger sample sizes and a balanced distribution of male and female participants. Third, we used a fixed single dose of 2g tyrosine per participant. While this makes the present supplementation regime comparable to a large number of previous studies [9], future studies could benefit from an examination of potential dose-response effects, and/or examine the potential benefits of individualized dosages. For example, van de Rest et al. (2017) [130] found that higher doses of tyrosine (amounting to 7-14g for a participant of 70 kg) were detrimental to WM task performance compared with lower doses. However, dose-response effects at lower amounts are unclear. Likewise, long-term effects of tyrosine supplementation have so far not been examined in detail, despite promising cognitive performance findings of habitual tyrosine intake in daily nutrition [11]. In addition, it could be argued that the tests on tyrosine-related changes in model parameters, such as directional Bayes Factors, should be considered for correction of multiple comparisons. However, in a hierarchical Bayesian framework, the risk of Type S (and likewise Type M) errors is substantially ameliorated as differences in posterior means are shrinked to a greater extent than posterior standard deviations in the light of partial pooling [131]. Furthermore, we did not observe a correlation between baseline spontaneous eye blink rate as a proxy measure of baseline DA levels [121] and tyrosine-related changes in task performance. However, future studies should consider possible modulatory effects of baseline DA or NA transmission levels [120,132], which will likely require much larger samples. Finally, although we assessed physiological arousal markers, we did not directly measure central DA or NA transmission nor related metabolites. Thus, direct evidence for increased catecholamine neurotransmission following the present supplementation regime (2g tyrosine) is still lacking. Nonetheless, several earlier studies in rodents and humans have shown that a single dose of tyrosine can increase catecholamine release [65,89,90].

## Conclusion

Potential cognitive enhancement effects of amino acid supplementation have gained considerable interest. Here we show that a single dose of tyrosine supplementation (2g) reduced decision thresholds across two value-based decision-making tasks, temporal discounting and model-based reinforcement learning, both of which have trans-diagnostic relevance. Model-based reinforcement learning was improved following tyrosine supplementation, and temporal discounting was, if anything, reduced. These findings, if replicated in a larger cohort, may be of particular significance for a range of (sub-)clinical psychiatric conditions. From a methodological perspective we show that model-based approaches can reveal novel insights into computational effects of supplementation. We also for the first time comprehensively report exploratory analyses of potential physiological correlates of tyrosine supplementation. These analyses revealed an overall reduction in physiological arousal following tyrosine supplementation, complementing previous approaches that focused solely on behavioral measures. Taken

together, these results suggest specific computational and physiological effects of tyrosine supplementation that future studies can build upon.

## Supporting information

**S1 Text. This SI file includes the following supporting informationSummary of posterior predictive checks of participants choices and response time distributions in both tasks (see S1 and S2 Figs); posterior distributions of *all* parameters from the DDM$_s$ of the RL task data (see S3 Fig); summary of the results from the softmax models of both tasks (see S4 and S5 Figs) and correspondence between task performance and physiology in more detail.**
(DOCX)

**S1 Fig. Posterior predictive checks for the winning DDM$_s$ and alternative model formulations for the seq. RL task data for the placebo and tyrosine condition.** In each plot the dotted line depicts participants' median RTs or mean choice behavior. Solid lines depict the median RTs and mean choice behavior drawn from 500 simulations of each of the different DDM formulations, as well as of a standard softmax model for choice data. The upper row depicts RT data and simulations and participants' probability to choose the same action as in the previous trial for the first decision stage S1 in relation to the options value differences ('decision conflict'). The lower row depicts S2 RT data and fraction of optimal choices in S2 (highest value option chosen) of participants and related model simulations.
(JPG)

**S2 Fig. Posterior predictive checks for the winning DDM$_s$ and alternative model formulations for the temporal discounting task data for the placebo and tyrosine condition.** In each plot the dotted line depicts participants' median RTs or mean % LL choices. Solid lines depict the median RTs or mean LL choices drawn from 500 simulations of each of the different DDM formulations, as well as of a standard softmax model for choice data. Data and simulations are plotted in relation to the absolute difference of LL (subjective values) and SS choice options.
(JPG)

**S3 Fig.** Posterior distributions of all group-level mean parameter from the DDM$_s$ for the seq. reinforcement learning data following placebo intake (left column, light grey plots) and their respective shifts related to tyrosine intake (right column, dark grey plots). Boxplots depict 80% and 90% HDIs. Depicted parameters are (A)-(C) learning rate $\eta_1$ and $\eta_2$ for prediction error related updating in S1and S2, and value decay rate of unchosen options $\gamma$, (D)-(F model-based parameter $\beta_{mb}$, model-free parameter $\beta_{mf}$ and drift-rate modulation by value differences in S2, (g) perseveration parameter $\rho$, (h)-(i) decision thresholds in stage S1 and S2 $\alpha_1$, $\alpha_2$, (j)-(k) non-decision times in S1 and S2 $\tau_1$, $\tau_2$, and (l)-(m) S1 and S2 drift-rate asymptotes $v_{1_{max}}$, $v_{2_{max}}$.
(JPG)

**S4 Fig.** Posterior distributions of the group-level means of the softmax model of the seq. RL task data following placebo intake (left column, light grey plots) and their respective shifts related to tyrosine intake (right column, dark grey plots). Boxplots depict 80% and 90% HDIs. Depicted parameters are from left to right: (A) learning-rate in S1 $\eta_1$, (B) learning-rate in S2 $\eta_2$, (C) decay rate of unchosen options $\gamma$, (D) model-based $\beta_{mb}$ weight and (E)(E) model-free $\beta_{mf}$ weight, (F S2 stage Q-value $\beta_2$ weight and (g) choice perseveration $\rho$.
(JPG)

**S5 Fig.** Posterior distributions of the group-level means of the softmax model of the temporal discounting task data following placebo intake (left column, light grey plots) and their

respective shifts related to tyrosine intake (right column, dark grey plots). Boxplots depict 80% and 90% HDIs. Depicted parameters are: (A) Discount rate log ($k$) and (B) softmax inverse temperature $\beta$.
(JPG)

**S6 Fig.** (A) Participants' pupil dilation at baseline (mean of t0 physiological measurements across tyrosine and placebo) was predictive of participants' average (S1 & S2) non-decision times ($\tau$) under placebo during seq ($r = -.5$, $p = .007$). RL. (B) Individual pupil dilation variability (PDV) at baseline was associated with the degree of drift-rate modulation by model-free Q-values ($\beta_{mf}$) during seq. RL under placebo ($r = .49$, $p = .009$). (C) Pupil dilation at baseline was also related to temporal discounting $\log(k)$ ($r = .51$, $p = .005$). Note, that the depicted associations fell short of significance after adjusting for False Discovery Rate (all p-values > FDR adjusted p-value = .003).
(JPG)

**S1 Table. Study sample characteristics (N = 28).** SE = standard error; YOE = years of education; BDI-II: Beck Depression Inventory-II; BIS-15: Barratt-Impulsiveness Scale (15 items); BIS/BAS: Behavioral Inhibition and Behavioral Activation System.
(DOCX)

**S2 Table. Effects on participants S1 RTs from a mixed effects regression analysis (rew = reward; trans = state transition; TYR = tyrosine).**
(DOCX)

**S3 Table. Model fit comparison of the DDMs in the temporal discounting & the seq. RL task via the Watanabe-Akaike Information Criterion (WAIC), the estimated log pointwise predictive density (elpd), and its difference to the winning model (DDMs).**
(DOCX)

**S4 Table. Proportions of correctly predicted binary choices (mean (range)) for the temporal discounting (TD) task data and for both stages (S1, S2) of the seq.** RL choice data, respectively. Values are computed via simulations based on 500 samples drawn from each of the respective single subject parameters' posterior distributions.
(DOCX)

## Author Contributions

**Conceptualization:** David Mathar, Jan Peters.

**Data curation:** David Mathar.

**Formal analysis:** David Mathar.

**Funding acquisition:** Jan Peters.

**Investigation:** Mani Erfanian Abdoust, Tobias Marrenbach.

**Methodology:** David Mathar, Deniz Tuzsus.

**Project administration:** Jan Peters.

**Supervision:** Jan Peters.

**Writing – original draft:** David Mathar.

**Writing – review & editing:** David Mathar, Deniz Tuzsus, Jan Peters.

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
