## [Decision Letter · Decision Letter 0]

5 Aug 2022

Dear Dr. Mathar,

Thank you very much for submitting your manuscript "The catecholamine precursor Tyrosine reduces autonomic arousal and decreases decision thresholds in reinforcement learning and temporal discounting" for consideration at PLOS Computational Biology.

As with all papers reviewed by the journal, your manuscript was reviewed by members of the editorial board and by several independent reviewers. In light of the reviews (below this email), we would like to invite the resubmission of a significantly-revised version that takes into account the reviewers' comments.

Notice especially the comments regarding better motivation for the study, and the need to make sure all data is available.

We cannot make any decision about publication until we have seen the revised manuscript and your response to the reviewers' comments. Your revised manuscript is also likely to be sent to reviewers for further evaluation.

Sincerely,

Ulrik R. Beierholm

Associate Editor

PLOS Computational Biology

Samuel Gershman

Deputy Editor

PLOS Computational Biology

Reviewer's Responses to Questions

**Comments to the Authors:**

Reviewer #1: Mathar and colleagues report a double-blind placebo-controlled study which examines the effect of the administration of L-tyrosine on two forms of value-based decision making: model-based RL and temporal discounting. Using a sophisticated form of model fitting that doesn’t take into account just choices, but also response times (hierarchical Bayesian DDM), they fit parameters of two computational models to the behavior on two tasks (the two-step task and a temporal discounting task) across both sessions, capturing L-tyrosine induced shifts in these parameters. The authors find that L-tyrosine reduced response times in both tasks, without observing decrements in performance. Consistent with these RT results, L-tyrosine attenuated the DDM thresholds in both tasks. Moreover, their modeling results also suggest that L-tyrosine increases the reliance on model-based control (and perseveration), and that it reduces the degree of temporal discounting. The authors also report a host physiological measures that predict temporal discounting.

I am of two minds about this paper. On the one hand, I think the research is clearly interesting and relevant, I appreciate the laudable rigor of the double-blind placebo-controlled design and the statistical analysis, and I think the results are interesting. I really only have a few very small comments to make there. However, I think that the paper lacks motivation and I think the discussion is not coherent.

In short, from the introduction it is not clear to me _why_ the authors decided to run this particular study. Of course, I understand that L-Tyrosine results in increases of dopaminergic availability, but given the previous literature, which found mixed findings regarding dopamine and the forms of decision making under investigation, it is not clear how this particular study will address this uncertainty. Is there a reason to believe that this form of supplementation more reliably increases dopamine? Is the double-blind placebo-controlled design particularly new? This is not clear.

As it stands now, the paper simply discusses two, relatively unrelated, forms of decision making sequentially, and then motivates the study through prior uncertainty. Aren’t there more theoretically interesting reasons to have run the study? What are some proposed theoretical roles for dopamine in model-based RL and temporal discounting? How would these results address these questions?

Moreover, I think the paper overpromises a combination of two tasks (line 100), but (as far as I understand) there is relatively little interaction between the tasks and the datasets they produce. The models are fit separately, and their results are interpreted separately. It’s unclear why the authors chose these particular tasks, and what they think they are getting by running them both in participants in the same session. Of course, efficiency and convenience are good reasons for doing this, but from the perspective of the paper it does not come across as particularly strong.

The discussion does not help clear this up. Analogous to the introduction, the results are discussed piece by piece, without drawing much attention to potential mutual mechanisms. The notable exception here is the discussion of RT and DDM threshold effects. However, most of the discussion reads as if the authors conduct two parallel studies that needed to be discussed in tandem. I also missed some more speculative theoretical discussion on the mechanisms underlying these L-tyrosine effects.

I think the authors should either considering making clearer what the benefit of the current setup is over prior studies (especially focusing on the use of these two tasks side-by-side). Alternatively, they can consider breaking up this one paper in two, so that they can pay more attention to the interesting but relatively separate literatures underlying theses tasks, without having to attempt to unite them under one banner.

Minor comments

I missed a discussion of Westbrook’s recent Science paper on dopamine and effort costs, which would nicely fit in with findings in both tasks.

Also, I saw a lot of papers from Colzato’s former group, and I want to make the authors aware that this PI has been undergoing a lot of scrutiny for research misconduct. If they did not know about, I think it would be worthwhile to read up on this situation.

Related to this latter point, there has recently been some very well justified criticism on the hypothesis that spontaneous EBR predicts dopamine availability. The authors can contribute to this burgeoning literature, but they do not cite one of its key players: https://www.ncbi.nlm.nih.gov/pmc/articles/PMC5602106/

For the modeling of the two-step task, I urge the authors to also model response key perseveration in addition to the stimulus perseveration (see Bolenz et al., 2019, eLife for an example).

It was a bit puzzling to see that the authors did not find evidence for increased model-based control in the L-tyrosine condition in the “model-agnostic” analyses. I expected the authors to address this point, but they did not. How is it possible that the effect is only found using the HDDM?

The authors claim that they changed the two-step task so that model-based RL yields increased reward rate. However, they do not directly demonstrate this. There is suggestive evidence, namely that reward rate is correlated with differences in RT between rare and common trials, but this is a distant proxy for model-based control. Why did they not demonstrate the advantage of model-based RL with simulations? Why did they not correlate decision making variables (such as the temperature parameter or the interaction effect on stay probabilities) with reward rate?

It appears to be the case that the effective learning rates for the two-step task model can become higher than 1 or lower than 0 (given the parameters that indicate the L-tyrosine induced shift). This seems problematic. How do the authors justify this theoretically? How often did these extreme cases occur?

Typos:

Equation on line 277 is wrong (unless everything right from Qmfs1 == 0), I think the left-most Q value should be indexed as t+1.

Line 303 misses closing parenthesis

Line 482: “proved to be a good test-retest reliability (Table1; Figure 1a)” 

Line 801-802: this sentence is not complete.

Reviewer #2: The authors present a well-written manuscript with solid experimental design and extensive modeling. I would like to ask the authors to respond to the following two points in a revision.

There is no effect of tyrosine on model-based control in terms of choices, as is evident from the model-agnostic analysis of stage 1 stay probabilities and the softmax model results. However, tyrosine is shown to affect reaction times. Most notably, tyrosine increases RT slowing on stage 2 following a rare transition; this might reflect some surprise that stems from participants’ awareness of the task structure (i.e., a model) and its transition probabilities. However, despite this more pronounced surprise response to a rare transition, tyrosine did not affect the reliance on these transition probabilities when deciding between stage 1 options. To what extent, then, can we conclude that tyrosine enhances model-based control rather than tyrosine enhancing the acquisition / awareness of task transition probabilities? That is, tyrosine seems to facilitate the acquisition of a task model but does not seem to promote the reliance on that model in choice behavior.

The posterior predictive checks in figure 1 of the supplementary material suggests that the models predict a rather linear increase in RT across low to medium-high decision conflict, whereas participants’ RTs is clearly nonlinear. To what extent is this a failure of the models to capture a key signature in the data? Can the authors elaborate on this discrepancy and how this might affect the results / conclusions?

Reviewer #3: Review uploaded as attachment

**Have the authors made all data and (if applicable) computational code underlying the findings in their manuscript fully available?**

Reviewer #1: None

Reviewer #2: Yes

Reviewer #3: **No: **Contrary to the data and code availability statement, modeling code is not available on OSF.

PLOS authors have the option to publish the peer review history of their article (what does this mean?). If published, this will include your full peer review and any attached files.

Reviewer #1: No

Reviewer #2: No

Reviewer #3: No
---

## [Decision Letter · Decision Letter 1]

28 Nov 2022

Dear Dr. Mathar,

Thank you very much for submitting your manuscript "The catecholamine precursor Tyrosine reduces autonomic arousal and decreases decision thresholds in reinforcement learning and temporal discounting" for consideration at PLOS Computational Biology. As with all papers reviewed by the journal, your manuscript was reviewed by members of the editorial board and by several independent reviewers. The reviewers appreciated the attention to an important topic. Based on the reviews, we are likely to accept this manuscript for publication, providing that you modify the manuscript according to the review recommendations.

Note that reviewer 3 has some minor suggestions. If you can make these changes quickly we are happy to approve the manuscript.

Sincerely,

Ulrik R. Beierholm

Academic Editor

PLOS Computational Biology

Samuel Gershman

Section Editor

PLOS Computational Biology

Reviewer's Responses to Questions

**Comments to the Authors:**

Reviewer #1: I am content with the authors' changes, and congratulate them on an interesting paper.

Reviewer #3: The authors have extensively revised the current manuscript and I highly appreciate their effort in replying to my comments on the original manuscript. The answers to my questions and remarks are comprehensible and clear. Overall, I find the revised manuscript to be stronger both from a theoretical as well as a methodological perspective.

Regarding my questions about the concrete realization of model-based control in the model, I was not aware about the algebraic equivalence. I thank the authors for pointing out this fact. Please add the respective paper cited in the methods section (Otto et al. 2013, PNAS) to the list of references.

In addition to my own remarks, I also read the comments from reviewer 1 and 2. I agree with both and found especially reviewer 1’s contributions to be insightful and relevant. Although I cannot speak for these reviewers, I found most of the authors’ replies to address the raised concerns adequately, further improving the revised manuscripts quality. Most notably, the revised manuscript now contains a clear rational/motivation for the study: tasks as trans-diagnostic markers and tyrosine as a possible supplement to improve function. However, in the discussion this topic is not further elaborated. Complementary to my colleague’s note on a lack of “some more speculative theoretical discussion”, the practical relevance of the current findings in light of clinical application (apart from Parkinson’s disease) is not clear from the discussion but might deserve some more explanation given the motivation stated in the introduction and the perpetual clinical endeavor.

**Have the authors made all data and (if applicable) computational code underlying the findings in their manuscript fully available?**

Reviewer #1: None

Reviewer #3: Yes

PLOS authors have the option to publish the peer review history of their article (what does this mean?). If published, this will include your full peer review and any attached files.

Reviewer #1: No

Reviewer #3: No

Figure Files:

Data Requirements:

Reproducibility:

References:

---

## [Editor Report · Decision Letter 2]

1 Dec 2022

Dear Dr. Mathar,

We are pleased to inform you that your manuscript 'The catecholamine precursor Tyrosine reduces autonomic arousal and decreases decision thresholds in reinforcement learning and temporal discounting' has been provisionally accepted for publication in PLOS Computational Biology.

Best regards,

Ulrik R. Beierholm

Academic Editor

PLOS Computational Biology

Samuel Gershman

Section Editor

PLOS Computational Biology

---

## [Editor Report · Acceptance letter]

19 Dec 2022

PCOMPBIOL-D-22-00800R2 

The catecholamine precursor Tyrosine reduces autonomic arousal and decreases decision thresholds in reinforcement learning and temporal discounting

Dear Dr Mathar,

I am pleased to inform you that your manuscript has been formally accepted for publication in PLOS Computational Biology. Your manuscript is now with our production department and you will be notified of the publication date in due course.

With kind regards,

Anita Estes
